# Flow Equivariant Recurrent Neural Networks

**T. Anderson Keller**
The Kempner Instutite for the Study of Natural and Artificial Intelligence
Harvard University, Cambridge, MA 15213
`t.anderson.keller@gmail.com`

## Abstract

Data arrives at our senses as a continuous stream, smoothly transforming from one instant to the next. These smooth transformations can be viewed as continuous symmetries of the environment that we inhabit, defining equivalence relations between stimuli over time. In machine learning, neural network architectures that respect symmetries of their data are called equivariant and have provable benefits in terms of generalization ability and sample efficiency. To date, however, equivariance has been considered only for static transformations and feed-forward networks, limiting its applicability to sequence models, such as recurrent neural networks (RNNs), and corresponding time-parameterized sequence transformations. In this work, we extend equivariant network theory to this regime of 'flows' – one-parameter Lie subgroups capturing natural transformations over time, such as visual motion. We begin by showing that standard RNNs are generally not flow equivariant: their hidden states fail to transform in a geometrically structured manner for moving stimuli. We then show how flow equivariance can be introduced, and demonstrate that these models significantly outperform their non-equivariant counterparts in terms of training speed, length generalization, and velocity generalization, on both next step prediction and sequence classification. We present this work as a first step towards building sequence models that respect the time-parameterized symmetries which govern the world around us.

## 1 Introduction

Every moment, the world around us shifts continuously. As you walk down the street, rooftops glide past, trees sway in the breeze, and cars drift in and out of view. These smooth, structured transformations – or flows – are not arbitrary, but instead are underpinned by a precise set of rules: an algebra which ensures self-consistency across space and time. Mathematically, these transformations which leave the essence of the world around you unchanged are called symmetries, and, in a sense, these symmetries over time can be seen to weave together the instantaneous into the continuous.

In machine learning, symmetry has emerged as a powerful handle from which we can grapple with the challenges of generalization and data efficiency. Models which encode symmetries are called 'equivariant', and their steady evolution has delivered significant performance improvements for data with known structure. Simultaneously, with the success of large language models, sequence modeling has become a prominent learning paradigm throughout machine learning sub-fields. We therefore ask: can these everyday smooth, continuous motion symmetries be encoded in our modern sequence models and leveraged for improved performance? Surprisingly, to date, the study of equivariance has largely been limited to temporally static transformations, narrowly missing the opportunity to capture the continuous time-parameterized symmetry transformations that dominate our natural experience (see Figure 1). In this work, we consider a carefully structured subset of natural motion transformations called flows: formally, one-parameter Lie subgroups. We introduce a formal notion of what it means for a sequence model to be 'flow equivariant' and proceed to demonstrate that existing sequence models are indeed *not* flow equivariant, *even* when all sub-components of these models are individually equivariant with respect to the full 'static' Lie group symmetry.

39th Conference on Neural Information Processing Systems (NeurIPS 2025).

Motivated by this gap in the literature, and the potential advantages of integrating a ubiquitous natural symmetry into sequence models, in this work, we study how to build Recurrent Neural Networks (RNNs) that are explicitly equivariant with respect to time-parameterized symmetry transformations. At the highest level, we demonstrate that we are able to do so in a manner very similar to Group equivariant Convolutional Neural Networks (G-CNNs): by lifting the hidden state of our RNN to a dimension of flows, indexed by the generating vector fields $\nu$. In slices, our model resembles a bank of RNNs (one for each flow), but weight sharing across $\nu$ turns that bank into a single group-equivariant dynamical system which guarantees generalization to unseen flows.

In the following, we will review the principle of equivariance in neural network architectures (§2), introduce the notion of 'Flow Equivariance' (§3), and demonstrate that existing RNNs are indeed not 'flow equivariant' as we would desire. We will then introduce Flow Equivariant Recurrent Neural Networks (FERNNs) in §4; and demonstrate empirically that FERNNs achieve zero-shot generalization to new flows at test time, improved length generalization, and improved performance on datasets which possess strong flow symmetries in §5. In our discussion, (§6), we briefly outline related work on equivariance with respect to motion, and highlight the differences with our proposed approach, but leave a thorough review of related work to §D. We conclude with the limitations of our proposed framework and promising future directions. We provide an accompanying blog post[1] with additional visualizations.

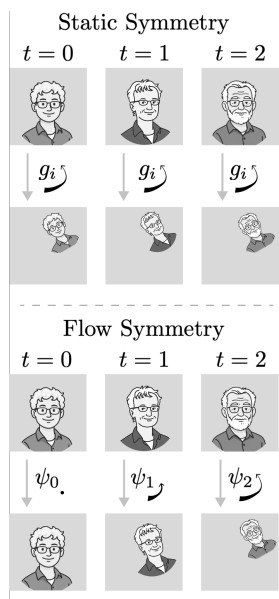

Figure 1: **Static vs. Flow Symmetries.** Static symmetries are constant while flows progress over time.

## 2 Equivariant Neural Networks

Informally, a network is said to be equivariant with respect to a set of structured transformations if every transformation of the input has a corresponding structured transformation of the output. The fact that this set of transformations is 'structured' is important. Typically, this structure is the abstract algebraic structure known as a 'group' $G$, meaning that any two transformations $g_i, g_j \in G$ will combine through a binary operation ($\cdot$) to form another known transformation $g_i \cdot g_j = g_k \in G$ (closure), and the 'chunking' of transformations is arbitrary, i.e. $(g_i \cdot g_j) \cdot g_k = g_i \cdot (g_j \cdot g_k)$ (associativity). As we will see later, we will exploit both these properties when designing equivariant models. At an intuitive level, an equivariant neural network $\Phi$ is a network whose activations 'live on the group'. This means that when $\Phi$ processes a transformed input, by associativity, this is equivalent to transforming the activations, and by closure, this transformation just yields another set of activations on the group – i.e., a structured transformation of the output.

Formally, a neural network layer $\Phi$ is equivariant with respect to a symmetry group $G$ if the action of the group commutes with the application of the layer. Following prior work [Cohen and Welling, 2016a], we will define this layer as a map between function spaces $\Phi : \mathcal{F}_K(X) \to \mathcal{F}_{K'}(Y)$, where $\mathcal{F}_K(X) := \{f : X \to \mathbb{R}^K\}$ denotes the set of signals defined on the abstract domain $X$ with $K$ channels. We will use the notation $\Phi[f]$ to denote the output of the model (in $\mathcal{F}_{K'}(Y)$ space) for an input signal $f$, and $\Phi[f](y) \in \mathbb{R}^{K'}$ to similarly denote the value of that output function at location $y$. For $f \in \mathcal{F}_K(X)$, we can then define the action of the group element $g \in G$ to be the left action:

$$(g \cdot f)(x) := f(g^{-1} \cdot x), \quad g \in G, \ x \in X. \tag{1}$$

Concretely, if $f$ is a 2D image, $X = \mathbb{Z}^2$ the set of pixel coordinates, and $G = (\mathbb{Z}^2, +)$ is the 2D translation group, then the translation action is: $(g \cdot f)(x) = f(x - g)$, a leftward shift of pixel coordinates. In the output domain, the action of $g$ depends on how we construct the equivariant map; but as we will show next, for group equivariant convolutional layers, this ends up being the same as it as for input, i.e. $(g \cdot \Phi[f])(y) = \Phi[f](g^{-1} \cdot y)$. Finally, the equivariance condition is then written as:

$$g \cdot \Phi[f] = \Phi[g \cdot f] \quad \forall \, g \in G, \ f \in \mathcal{F}_K(X) \tag{2}$$

We can see this is the formalization of our intuitive definition: the action of the group on the input $f$ can instead be pulled through to act on the lifted group indices of the network's output $\Phi[f]$.

---

[1]https://kempnerinstitute.harvard.edu/research/deeper-learning/flow-equivariant-recurrent-neural-networks/

**Group Convolution.** A simple example of such an equivariant map $\Phi$ is the group convolution [Cohen and Welling, 2016a]: a generalization of the standard convolutional neural network (CNN) layer from the translation group to other arbitrary discrete groups. To accomplish this, we define kernels to be functions on the group $G$, i.e. $\mathcal{W}^i : G \to \mathbb{R}^K$, and perform 'lifting' with kernels $\mathcal{U}^i : X \to \mathbb{R}^K$ at the first layer of the network to 'lift' from the input space to the group space. Mathematically, we write the lifting convolution ($\hat{\star}_G$) and subsequent group convolutions ($\star_G$) as:

$$[f \hat{\star}_G \mathcal{U}^i](g) = \sum_{x \in \mathbb{Z}^2} \sum_{k=1}^{K} f_k(x) \mathcal{U}_k^i(g^{-1} \cdot x) \quad \& \quad [f \star_G \mathcal{W}^i](g) = \sum_{h \in G} \sum_{k=1}^{K} f_k(h) \mathcal{W}_k^i(g^{-1} \cdot h) \quad (3)$$

Where $i$ and $k$ are the output and input channel indices of the convolution respectively. We see that the output of the layer is now defined over the group $G$, and we can verify that these layers are indeed equivariant with respect to Equation 2 (see §A.1). The important property being that $[(g \cdot f) \star_G \mathcal{W}^i] = g \cdot [f \star_G \mathcal{W}^i]$, that is, the convolution commutes with the group action.

**Group Equivariant Recurrent Neural Networks.** In this work we define a recurrent neural network $h$ as a map between space-time function spaces $h : \mathcal{F}_K(X, \mathbb{Z}) \to \mathcal{F}_{K'}(Y, \mathbb{Z})$, where $\mathcal{F}_K(X, \mathbb{Z}) := \{f : X \times \mathbb{Z} \to \mathbb{R}^K\}$ denotes the set of time-varying $K$-dimensional signals defined over space $X$ and (discretized) time $t \in \mathbb{Z}$. In the remainder of this paper we will use discrete time domains for our space-time signals, but we note that in order to accommodate continuous time sequence models, time may be made continuous ($t \in \mathbb{R}$) by changing domain of $h$, with only minor changes to the analysis. We can then define a group-equivariant RNN (G-RNN) as a simple RNN with group convolutions in place of both the usual linear recurrence and input maps. Explicitly:

$$h_{t+1}^i[f_{\leq t}] = \sigma\big(h_t[f_{<t}] \star_G \mathcal{W}^i + f_t \hat{\star}_G \mathcal{U}^i\big), \tag{4}$$

where $\mathcal{W}^i : G \to \mathbb{R}^K$, and $\mathcal{U}^i : G \to \mathbb{R}^K$ are the group convolutional kernels for the hidden-to-hidden mapping and input-to-hidden mapping respectively (for output channel $i$), and $h_0$ is initialized to be invariant to the group action, i.e. $h_0(g) = h_0(g') \; \forall g, g' \in G$. The notation $h_t[f_{<t}]$ implies that $h_t$ is a causal function of the input signal $f$ prior to time $t$. The non-linearity $\sigma$ is assumed group-equivariant as well (pointwise). In §A.2, we prove by induction that this network is indeed equivariant with respect to action of the group $G$ on the input signal $f$, meaning $h_t^i[g \cdot f_{\leq t}] = g \cdot h_t^i[f_{\leq t}] \; \forall g \in G, t \in \mathbb{Z}, f \in \mathcal{F}_K(X, \mathbb{Z})$. Intuitively, this is because both the input-to-hidden mapping and the recurrent mapping are equivariant, meaning any constant transformation of the input can be pulled through both mappings (and the nonlinearity) yielding a corresponding transformation of the output (hidden state). Concretely, this implies that a recurrent neural network with convolutional recurrent connections, and convolutional encoders is indeed still equivariant with respect to a static translation of the entire input sequence. However, as we will show next, *this does not imply that the RNN will be equivariant with respect to time-parameterized translation, i.e. motion.*

## 3 Flow Equivariance

Like the pathline traced out by a droplet of dye carried along by a moving fluid, the flow $\psi_t(\nu)$ generated by a vector field $\nu$ gives the position of any point after time $t \in \mathbb{R}$. More formally, a flow is a one-parameter subgroup of a Lie group $G$, generated by an element $\nu$ of the corresponding Lie algebra $\mathfrak{g}$ of $G$ [Hall, 2015]. We denote a generator by $\nu \in \mathfrak{g}$, and its time-$t$ flow $\psi_t(\nu) : \mathbb{R} \times \mathfrak{g} \to G$ by $\psi_t(\nu) = \exp(t\nu) \in G$, where $\exp$ is the exponential map. By construction, the flow admits an identity at $t = 0$, $\psi_0(\nu) = e$, and a composition property: $\psi_s(\nu) \cdot \psi_t(\nu) = \psi_{s+t}(\nu)$, such that flowing for time $s$ and then $t$ is the same as flowing for time $s + t$. Intuitively, the flow $\psi_t(\nu)$ is the group element that transports a point from the identity to where it will be after time $t$ when moving with instantaneous velocity $\nu$. Colloquially, we therefore call $\psi$ a *time-parameterized symmetry transformation*. For a given $f \in \mathcal{F}_K(X, \mathbb{Z})$ and generator $\nu \in \mathfrak{g}$, we define a flow acting on $f$ as

$$(\psi(\nu) \cdot f)_t(x) := f_t(\psi_t(\nu)^{-1} \cdot x). \tag{5}$$

and identically for a flow acting on a function in the output space $\mathcal{F}_{K'}(Y, \mathbb{Z})$: $(\psi(\nu) \cdot \Phi[f])_t(y) := \Phi_t[f](\psi_t(\nu)^{-1} \cdot y)$. Throughout this work we will use the notation $(\psi(\nu) \cdot f)$ and $(\psi(\nu) \cdot \Phi)$, leaving time implicit, to denote the flow acting on the entire time domain of the function simultaneously. We can then define 'flow equivariance' as the property of a sequence model such that its sequential output commutes with such time-parameterized transformations of the input:

**Definition 3.1** (**Flow Equivariance**). For a set of generators $V \subseteq \mathfrak{g}$, a sequence model $\Phi : \mathcal{F}_K(X, \mathbb{Z}) \to \mathcal{F}_{K'}(Y, \mathbb{Z})$ is *equivariant with respect to the flows generated by V* iff

$$\psi(\nu) \cdot \Phi[f] = \Phi[\psi(\nu) \cdot f] \quad \forall \nu \in V, \ f \in \mathcal{F}_K(X, \mathbb{Z}). \tag{6}$$

**Flow Equivariance of Sequence Models.** Our proposed Definition 3.1 has a noteworthy special case which captures much of the literature to date on equivariance in sequence models. Specifically:

*Remark* (Frame-wise feed-forward equivariant models are trivially flow equivariant.). Let $\Phi[f] = [\phi(f_0), \phi(f_1) \dots \phi(f_T)]$ be a frame-wise applied equivariant map, with $\phi : \mathcal{F}_K(X) \to \mathcal{F}_{K'}(Y)$ such that: $\phi(\psi_t(\nu) \cdot f_t) = \psi_t(\nu) \cdot \phi(f_t) \ \forall t \in \mathbb{Z}, \ \nu \in V$. Then $\Phi$ is flow equivariant by Definition 3.1.

See §A.3 for the short proof. Intuitively, one sees this very clearly for CNNs applied to video sequences frame-wise: since the CNN commutes with the transformation of each frame, and the transformation on the sequence distributes amongst the frames (by linearity of the action), the act of transforming the input frame-by-frame is equivalent to transforming the output frame-by-frame. Since this is achieved through a complete lack of sequence modeling in $\Phi$, we call it 'trivial flow equivariance'. Equivalently, one can see that the G-RNN defined in Equation 4 with $\mathcal{W} = \mathbf{0}$ reduces to such a feed-forward equivariant map (since recurrence is removed), and is thus also trivially flow equivariant. To achieve more sophisticated flow equivariance in the context of sequence processing, we must use sequence models with non-zero recurrence. Notably, while in §2 we show that group-equivariant RNNs do commute with static transformations, we show below that such models generally do not commute with time-parameterized transformations, meaning they are not flow equivariant.

*Theorem* 3.1 (G-RNNs are not flow equivariant). A G-RNN as defined in Equation 4, with non-zero $\mathcal{W}$, is not flow equivariant according to Definition 3.1, except in the degenerate flow-invariant case.

*Proof Sketch.* (Theorem 3.1) As visualized in Figure 2, let $\sigma = \mathrm{Id}$, and $[* \ \hat{\star}_G \mathcal{U}]$, $[* \star_G \mathcal{W}]$ be the identity, such that the hidden state evolves simply as a sum of the past state and the new input: $h_{t+1}[f_{\leq t}] = h_t[f_{<t}] + f_t$. If the input is a static bump centered at 0, $h_{t+1}[f_{\leq t}]$ will be an ever-growing bump, while $h_{t+1}[\psi(\nu) \cdot f_{\leq t}]$ will be a 'train' of bumps, since, at each time step, the hidden state is 'lagging behind' the input which has shifted by $\psi_1(\nu)$. Thus a corresponding shift of the growing bump will never equal the train of bumps, i.e. $(\psi(\nu) \cdot h[f_{\leq t}])_{t+1} \neq h_{t+1}[\psi(\nu) \cdot f_{\leq t}]$. $\square$

*Remark.* (Degenerate flow invariance) If kernels $\mathcal{W}$ & $\mathcal{U}$ are constant on $G$, then the outputs of Equation 4 are spatially uniform. Since the flow action permutes only the G-index, these constants are fixed points, making the G-RNN flow invariant.

One can verify that this counterexample applies to most other existing sequence models, including state space models, gated recurrent neural networks, neural ODEs, and even transformers. In the following section, we show how to build a recurrent neural network which *is* equivariant with respect to these transformations, and argue that this same construction can likely be carried over to other sequence models as well.

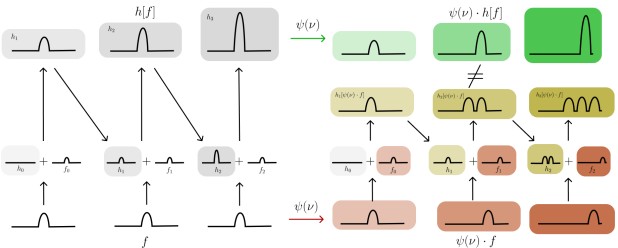

Figure 2: **G-RNNs are not generally flow equivariant.** We show this by simple counterexample with: $h_{t+1} = h_t + f_t$. See §A.4 for the full proof.

## 4 Flow Equivariant Recurrent Neural Networks

In this section we introduce our Flow Equivariant Recurrent Neural Network (FERNN) and prove the equivariance condition (Definition 3.1) holds. Inspired by the above analysis of the failure of G-RNNs to be flow equivariant, we note that a simple condition by which equivariance can be achieved is if computation is performed 'in the reference frame' of the signal. For the case of static symmetry groups, we can interpret the group convolution as performing the same operation (dot product of filter and signal) in all possible statically transformed reference frames (group elements). In the case of time-parameterized symmetry groups, we can then assume that we desire to be performing the same recurrent update in all possible dynamically transformed (moving) reference frames. To achieve this,

we see that, analogous to the $g$-lift in the G-CNN, we must now lift our hidden state and input to the flow elements $\nu \in V$. To do so, we define the action of the instantaneous group elements $\psi_t(\nu) \in G$ on the lifted $V \times G$ space to be the same left action as before, only on the $G$ dimension. Explicitly, for $h_t(\nu, g) : V \times G \to \mathbb{R}^{K'}$, we define $\psi_t(\nu) \cdot h_t(\nu, g) := h_t(\nu, \psi_t(\nu)^{-1} \cdot g)$.

**Input Flow-Lifting Convolution.** To lift the input to the product group $V \times G$, we will re-use the original lifting group convolution from Equation 3 exactly, and simply perform a 'trivial lift' to the additional $V$ dimension by copying the output of the convolution for each value of $\nu$. Explicitly:

$$[f \;\hat{\star}_{V \times G}\; \mathcal{U}^i](\nu, g) = \sum_{x \in X} \sum_{k=1}^{K} f_k(x) \mathcal{U}_k^i(g^{-1} \cdot x) \tag{7}$$

In §A.5 we prove that this is equivariant with respect to the group action, yielding the following:

$$\psi_t(\nu) \cdot [f_t \;\hat{\star}_{V \times G}\; \mathcal{U}^i](\nu, g) = [(\psi_t(\nu) \cdot f_t) \;\hat{\star}_{V \times G}\; \mathcal{U}^i](\nu, g) := [f_t \;\hat{\star}_{V \times G}\; \mathcal{U}^i](\nu, \psi_t(\nu)^{-1} \cdot g) \tag{8}$$

We note that it is also possible to perform a non-trivial lift, where $\mathcal{U}$ is defined on the lifted space $V \times G$, as in G-CNNs. This changes the later recurrence relation (Eqn. 11) necessary to achieve flow equivariance, but is largely a design choice. In the experiments and proofs we stick with the trivial lift, but describe the model with non-trivial lift in §C.1.

**Flow Convolution.** The group convolution can then be extended to our lifted hidden state as:

$$[h \star_{V \times G} \mathcal{W}^i](\nu, g) = \sum_{\gamma \in V} \sum_{m \in G} \sum_{k=1}^{K'} h_k(\gamma, m) \, \mathcal{W}_k^i(\gamma - \nu, g^{-1} \cdot m). \tag{9}$$

We see that for the convolution over $V$, we have assumed that $V \subseteq \mathfrak{g}$ is an additive (abelian) discrete group, such that the combination $(-\nu) \cdot \gamma = \gamma - \nu$. If $V$ is not abelian, the general Baker-Campbell-Hausdorff product $((-\nu) \cdot \gamma)$ should be used instead [Hall, 2015]. Similarly, if $V$ is not discrete, one should perform an integral with respect to the Haar measure. In all experiments we use abelian $V$, and a discrete grid for $V$, so this is exact. In §A.6 we prove this is equivariant with respect to the action of individual flow elements. Explicitly:

$$[(\psi_t(\nu) \cdot h) \star_{V \times G} \mathcal{W}^i](\nu, g) = \psi_t(\nu) \cdot [h \star_{V \times G} \mathcal{W}^i](\nu, g) \tag{10}$$

Intuitively, this is simply because the individual group elements along the flow only act on the $G$ coordinate of the signal, and thereby commute with the standard group convolution.

**Flow Equivariant Recurrence Relation.** We define the recurrence relation for the FERNN as:

$$h_{t+1}(\nu, g) = \sigma\big(\psi_1(\nu) \cdot [h_t \star_{V \times G} \mathcal{W}](\nu, g) + [f_t \;\hat{\star}_{V \times G}\; \mathcal{U}](\nu, g)\big). \tag{11}$$

We see that there are two primary differences between this model and the G-RNN: the extra dimension $\nu$, and the action of the instantaneous one-step flow element $\psi_1(\nu)$ at each position $\nu$ of the lifted hidden state. Intuitively, this can be understood as a bank of G-RNNs, each flowing autonomously according to their corresponding vector fields $\nu$ simultaneously. Similar to G-CNNs then, when an input arrives undergoing a given flow transformation with generator $\hat{\nu}$, this construction allows us to 'pull out' this transformation into a shift of the $\nu$ dimension. Formally:

*Theorem* 4.1. (FERNNs are flow equivariant) Let $h[f] \in \mathcal{F}_{K'}(Y, \mathbb{Z})$ be a FERNN as defined in Equations 7, 9, and 11, with hidden-state initialization invariant to the group action and constant in the flow dimension, i.e. $h_0(\nu, g) = h_0(\nu', g) \;\forall \nu', \nu \in V$ and $\psi_1(\nu) \cdot h_0(\nu, g) = h_0(\nu, g) \;\forall \nu \in V, g \in G$. Then, $h[f]$ is flow equivariant according to Definition 3.1 with the following re-defined representation of the action of the flow in the output space:

$$(\psi(\hat{\nu}) \cdot h[f])_t(\nu, g) = h_t[f](\nu - \hat{\nu}, \psi_{t-1}(\hat{\nu})^{-1} \cdot g) \tag{12}$$

In words, the equivariance condition $h_t[\psi(\hat{\nu}) \cdot f](\nu, g) = h_t[f](\nu - \hat{\nu}, \psi_{t-1}(\hat{\nu})^{-1} \cdot g)$ means that the hidden state of the FERNN processing a flowing input sequence has a corresponding flow along the $G$ dimension, and a shift along the $V$ dimension, both determined by the input flow generator $\hat{\nu}$. In §A.7 we prove Theorem 4.1 by induction, and give a brief sketch below.

*Proof Sketch.* (Theorem 4.1) We wish to prove $h_t[\psi(\hat{\nu}) \cdot f](\nu, g) = h_t[f](\nu - \hat{\nu}, \psi_{t-1}(\hat{\nu})^{-1} \cdot g) \;\; \forall t$. Letting $H = V \times G$, we write the recurrence relation for the transformed input as:

$$h_{t+1}[\psi(\hat{\nu}) \cdot f](\nu, g) = \sigma\big(\psi_1(\nu) \cdot [h_t[\psi(\hat{\nu}) \cdot f_{<t}] \star_H \mathcal{W}](\nu, g) + [(\psi_t(\hat{\nu}) \cdot f_t) \,\hat{\star}_H\, \mathcal{U}](\nu, g)\big). \quad (13)$$

Assume we initialize the hidden state to a fixed point of the flow-element action $\psi_1(\nu) \cdot h_0(\nu, g) = h_0(\nu, g)$, constant in $\nu$. We can see that the base case is satisfied trivially since $\psi_1(\nu) \cdot h_0(\nu, g) = \psi_{-1}(\nu) \cdot h_0(\nu - \hat{\nu}, g)$ by constancy. We then assert the inductive hypothesis, allowing us to substitute $h_t[\psi(\hat{\nu}) \cdot f](\nu, g) \Rightarrow h_t[f](\nu - \hat{\nu}, \psi_{t-1}(\hat{\nu})^{-1} \cdot g) = \psi_{t-1}(\hat{\nu}) \cdot h_t[f](\nu - \hat{\nu}, g)$ into equation 13. Similarly, due to the trivial lift and the equivariance of lifting convolution, we can substitute $[(\psi_t(\hat{\nu}) \cdot f_t) \,\hat{\star}_{V \times G}\, \mathcal{U}](\nu, g) \Rightarrow \psi_t(\hat{\nu}) \cdot [f_t \,\hat{\star}_{V \times G}\, \mathcal{U}](\nu - \hat{\nu}, g)$. Ultimately, this gives us:

$$(13) = \sigma\big(\psi_1(\nu) \cdot \psi_{t-1}(\hat{\nu}) \cdot [h_t[f_{<t}] \star_H \mathcal{W}](\nu - \hat{\nu}, g) + \psi_t(\hat{\nu}) \cdot [f_t \,\hat{\star}_H\, \mathcal{U}](\nu - \hat{\nu}, g)\big). \quad (14)$$

By the properties of flows, we know that $\psi_1(\nu) \cdot \psi_{t-1}(\hat{\nu}) = \psi_1(\nu) \cdot \psi_{-1}(\hat{\nu}) \cdot \psi_t(\hat{\nu}) = \psi_t(\hat{\nu}) \cdot \psi_1(\nu - \hat{\nu})$, and thus we can pull out a factor of $\psi_t(\hat{\nu})$ from both terms, leaving us with the original recurrence defined over a new $V$ index $\nu - \hat{\nu}$, exactly the shift in $V$. $\qquad\square$

Intuitively, the idea is that we would like to build a sequence model which fixes the lack of equivariance demonstrated by the counterexample in Figure 2. To accomplish this, we note that the hidden state of the model in Figure 2 appears to be lagging behind the input at each time step. To fix this, we therefore propose to augment our hidden state with a bank of flows, each generated by a separate vector field $\nu$ such that when the input arrives at a specific velocity, it will exactly match one of these flow parameters, effectively shifting the 'zero' point of the moving coordinates of our RNN to this new flow. Since these flows are relative quantities (obeying the group law), we see that all the other $\nu$ channels in the bank will similarly shift according to their relative position in $V$ space (e.g. relative velocity). These effects combined lead to the output representation we observe analytically. In Figure 9 we visualize the model and the same example of a moving bump from Figure 2, providing a clear demonstration of what the action of the flow on the hidden state looks like in practice.

We finally note that since the hidden state lives on the product group $V \times G$, and the input is similarly lifted to the product group, one can simply stack these layers, replacing the lifting convolution with another flow convolution and maintain flow equivariance. Since a composition of equivariant maps is itself an equivariant map [Kondor and Trivedi, 2018], a deep FERNN is also flow equivariant (provided all non-linearities, normalization layers, and other additional architectural components are additionally flow equivariant – we refer readers to [Bronstein et al., 2021] for an extensive review).

**Practical Implementation.** We note that exact equivariance requires that the group $V$ be closed under the action of the flow on the output. In theory, this can be satisfied by many constructions, but in practice, we cannot lift our hidden state to an infinite dimensional group. Therefore, similar to prior work [Worrall and Welling, 2019], we implement a truncated version of the group $V$, and incur equivariance error at the boundaries. Furthermore, in the majority of experiments in this work, we define the flow convolution to not mix the $V$ channels at all, explicitly: $\mathcal{W}_k^i(\nu, g) = \delta_{\nu=e} \mathcal{W}_k^i(g)$. We find that this is beneficial to not propagate errors between the $\nu$ channels induced by the truncation, similar to prior work on scale equivariance [Sosnovik et al., 2020b]. In Appendix §E.5 we demonstrate cases where including $V$-mixing helps modeling inter-velocity interactions (e.g. object collisions).

## 5 Experiments

In the following, we introduce datasets with known flow symmetries and study how adding flow equivariance impacts performance compared with non-equivariant baselines. We investigate two sequence datasets: (i) next-step prediction on a modified 'Flowing MNIST' dataset with 2 simultaneous digits undergoing imposed translation and rotation flows [LeCun et al., 1998], and (ii) sequence classification on the KTH human action recognition dataset [Schuldt et al., 2004], augmented with additional translation flows to simulate camera motion. In §B we include the full details of the dataset creation, model architectures, training, and evaluation procedures; and in §E we include extended results. Code is available at: `https://github.com/akandykeller/FERNN`.

**Flowing MNIST Datasets.** We construct a number of variants of the Flowing MNIST dataset to test generalization, each defined by the set of generators we use for training and evaluation ($V_{train}$, $V_{test}$). In words, sequences are composed of two digits moving with separate random translation or rotation

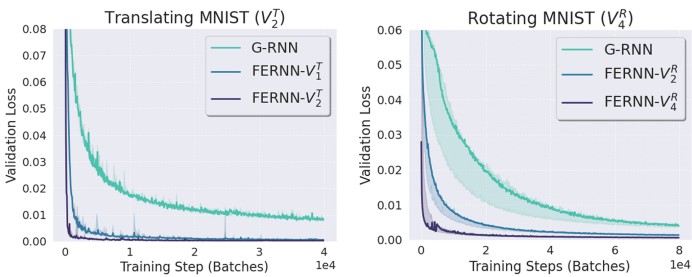

| Model | Test MSE ↓ |
|---|---|
| $V_{train} = V_{test} = V_2^T$ | |
| G-RNN | $8.1e{-3} \pm 6e{-4}$ |
| FERNN-$V_1^T$ | $5.3e{-4} \pm 8e{-5}$ |
| FERNN-$V_2^T$ | $\mathbf{1.5e{-4} \pm 2e{-5}}$ |
| $V_{train} = V_{test} = V_4^R$ | |
| G-RNN | $4.0e{-3} \pm 5e{-4}$ |
| FERNN-$V_2^R$ | $1.3e{-3} \pm 5e{-5}$ |
| FERNN-$V_4^R$ | $\mathbf{6.1e{-4} \pm 3e{-5}}$ |

Figure 3: **Increased flow equivariance increases training speed on data with flow symmetry**. Validation loss vs. train steps.

Table 1: **Flowing MNIST Test MSE.** (mean $\pm$ std, 5 seeds)

velocities. For rotation flows, we discretize the rotation algebra $\mathfrak{so}(2)$ at $\Delta\theta = 10^o$ intervals, yielding: $V_N^R = \{k\Delta\theta J | k = -N, \ldots N\}$, where $J = \left(\begin{smallmatrix} 0 & -1 \\ 1 & 0 \end{smallmatrix}\right) \in \mathfrak{so}(2)$. For 2D translation flows, we always use the integer lattice up to velocities $N\frac{pixels}{step}$: $V_N^T = \{\nu \in \mathbb{Z}^2 \mid ||\nu||_\infty \leq N\}$. See Figures 4 & 5 for examples. A train/test example from the dataset is a sequence $f_t$ constructed by drawing two random static samples $f^1, f^2$ from the original MNIST dataset and two corresponding random vector fields $\hat{\nu}^1, \hat{\nu}^2$ uniformly from the corresponding $V_*$, applying the action of the corresponding flows to each sample, and summing the results to generate the timeseries: $f_t = \psi_t(\hat{\nu}^1) \cdot f^1 + \psi_t(\hat{\nu}^2) \cdot f^2$.

**Next-Step Prediction Task.** Models are trained to solve an autoregressive forward prediction task: given the first $T = 10$ time-steps of the timeseries $f$, predict the next 10 time-steps. Model parameters are optimized to minimize the mean-squared error (MSE) between predictions and the ground truth for 50 epochs. In the length generalization experiments of Figure 4, we evaluate the model's MSE on forward prediction up to 70 steps, significantly longer than the 10 seen in training.

**Next-Step Prediction Models.** We use G-RNNs and FERNNs exactly as defined in Equations 4 & 11 respectively, where $G$ is defined as $SO(2)$ with translations ($SE(2)$) for the rotation flow datasets, and $G = (\mathbb{Z}^2, +)$ is the standard 2D translation group for the translation flow datasets. We use the *escnn* library to implement the $SE(2)$ convolutions [Cesa et al., 2022]. We set $\sigma = \text{ReLU}$, use a single recurrent layer, a hidden state size $K'$ of 128 channels, and $(3 \times 3)$ convolutional kernels in the spatial dimensions. We note that the spatial dimensions of the input are preserved throughout the entire forward pass, i.e. there is no spatial pooling, stride, or linear layers. We augment these models with a small 4-layer CNN decoder $g_\theta(h_{t+1})$ with 128 channels, and ReLU activations, which maps from each hidden state to the corresponding output. Explicitly: $g_\theta(h_{t+1}) = \hat{f}_{t+1}$, where the training loss is the MSE between $\hat{f}_{t+1}$ and $f_{t+1}$ averaged over time $t \in [11, 20]$. For the FERNN models, we also crucially 'max-pool' over the $V$ dimension of the hidden state before decoding, e.g. $g_\theta(\max_\nu h_{t+1}(\nu, g)) = \hat{f}_{t+1}$. This means that all G-RNN and FERNN models have exactly the same number of parameters, since the FERNN can be seen to share its parameters over the different flow elements. For each FERNN, we denote its corresponding set of generators with FERNN-$V_N^*$.

**Next-Step Performance.** In Figure 3 we show the validation loss curves for these models on Flowing MNIST, and in Table 1 we show the final MSE on the test sets (mean $\pm$ std. over 5 random initializations). We see that on both datasets, FERNNs dramatically outperform G-RNNs, reaching nearly an order of magnitude lower error, and doing so in significantly fewer iterations. We also see that even partial equivariance (e.g. FERNN-$V_1^T$ on $V_2^T$) is highly beneficial to model performance.

**Length Generalization** In Figure 4 we plot generated forward predictions on Translation-Flow MNIST with $V_{train} = V_{test} = V_2^T$ for both the G-RNN and FERNN-$V_2^T$ models compared with the ground truth sequence. The length of sequences seen in training is indicated by the shaded gray area, and at test time we autoregressively generate sequences from the models which are significantly longer than this. We see that the G-RNN predictions begin to degrade after the training length, while the FERNN remains highly consistent. We quantify this error for each forward prediction step T in the adjacent plot, demonstrating that indeed FERNNs dramatically outperform G-RNNs in length generalization. We note that the performance improvement of FERNNs over G-RNNs is more significant for translation flows compared with rotation flows in our experiments. Particularly, we do not see quite as strong length generalization on Rotating MNIST (see §E); however, we speculate

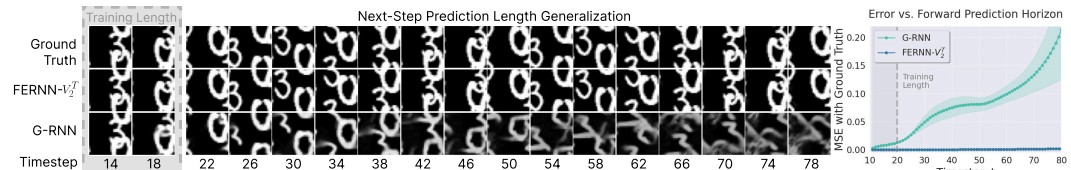

Figure 4: **FERNNs exhibit next-step prediction length-generalization capabilities far surpassing G-RNNs on simple flows**. We plot samples for the forward prediction trajectories of a G-RNN and FERNN-$V_2^T$ trained on Translating MNIST $V_2^T$ to predict 10-steps into the future (down-sampled by a factor of 4 in time for visualization). We see the G-RNN performs well on this training regime but diverges rapidly with lengths longer than training. The FERNN generalizes nearly perfectly. On the right, we plot this forward prediction error vs. the forward prediction timestep for both models.

that this is due to the interpolation procedure necessary to implement small rotations on a finite grid, not due to any difference in the underlying theory. Interpolation of rotating pixel grids is known to break exact equivariance, and we suggest prior methods for addressing this issue in G-CNNs may be equally applicable to FERNNs [Diaconu and Worrall, 2019].

**Velocity Generalization**  Similar to standard group equivariant neural networks, weight sharing induces automatic generalization to previously unseen group elements. In the case of flow equivariance, we then expect zero-shot generalization to flow generators that are part of the lifted hidden state's $V$ but not seen during training. In Figure 5, we show example sequences which demonstrate this to be the case. Specifically, the models are trained on low velocity examples ($V_1^T$ and $V_2^R$), and evaluated on the full set of higher velocity generators (from $V_2^T$ and $V_5^R$ respectively). In the adjacent plots, we show the test MSE for sequences flowing according to each generator. We see that the FERNNs (middle) generalize nearly perfectly, while the G-RNNs (bottom) only learn to forward predict in-domain flows, and fail to generalize outside. Furthermore, in Figure 6, we evaluate the performance of FERNNs trained and tested on velocities outside of their integrated $V$ set. We see that while error increases slightly, FERNNs can generally interpolate between integrated velocity values to yield significantly improved performance on a wider range of data.

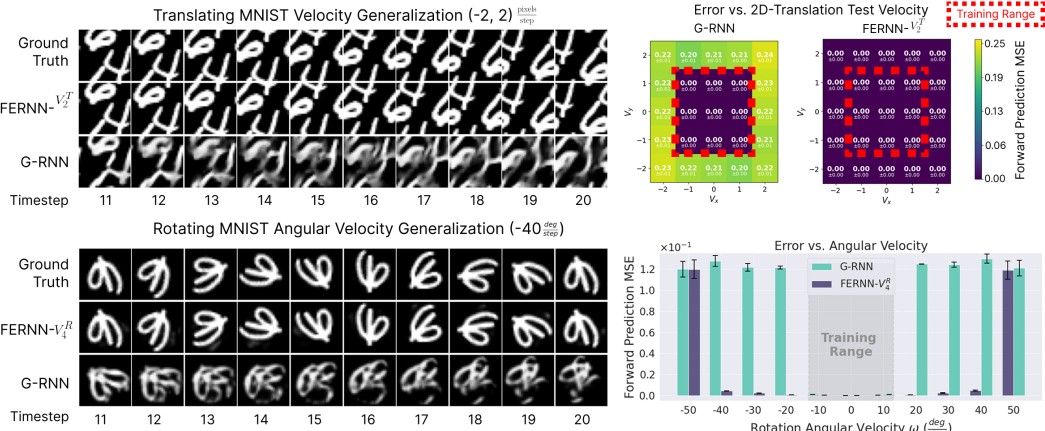

Figure 5: **FERNNs exhibit next-step prediction generalization to previously unseen translation and rotation flow velocities where G-RNNs fail**. Samples of forward predictions for FERNNs ($V_2^T$ & $V_4^R$) trained on Translation & Rotating MNIST with limited flow velocities ($V_1^T$, $V_1^R$) and tested on a wider range of velocities ($V_2^T$ & $V_5^R$). We see the FERNNs achieve near perfect forward-prediction performance within their $V$ set. On the right we plot the full distribution of errors across $V_{test}$.

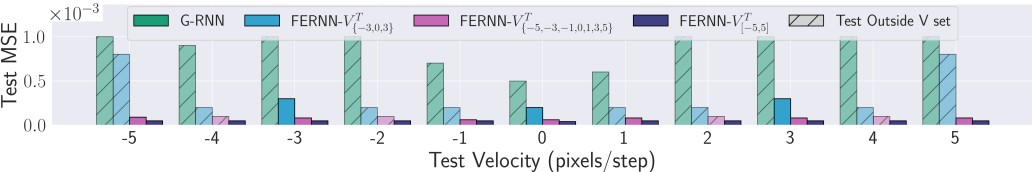

Figure 6: **FERNNs can interpolate between** $\nu \in V$. Test MSE for FERNNs trained and tested on $x$-translation velocity flows from $-5$ to $+5$, with varying $V$-set coverage (see legend). Hatched bars indicate the test velocity is outside the model's equivariant set.

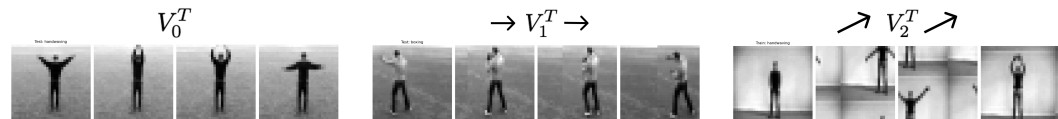

$V_0^T$        $\to V_1^T \to$        $\nearrow V_2^T \nearrow$

Figure 7: **Moving KTH Dataset Samples**. Translation flow augmentation to simulate camera motion.

**Moving KTH Action Recognition.** Finally, we test our FERNN model's ability to classify natural video sequences with induced motion using the simple task of Action Recognition from the KTH dataset. The dataset is composed of videos of 25 people performing 6 different actions. We use the traditional train-val-test split, by person, requiring strong generalization to new individuals. We down-sample each clip by a factor of 2 in space and time for computational convenience, leading to grayscale sequences $f$ of size 32x32, with 16 time-steps each. In our experiments, we apply flows from $V_1^T$ and $V_2^T$ to the dataset as an attempt to emulate slow camera motion (see Figure 7). On this dataset, differently from MNIST, we do not re-sample velocities per example, leading to a more realistic restricted dataset. In Appendix §E.6, we include results with this velocity data augmentation.

| Model | Test Acc. ↑ | # $\theta$ |
|---|---|---|
| 3D-CNN | $0.626 \pm 0.02$ | 209K |
| G-RNN+ | $0.639 \pm 0.02$ | 242K |
| G-RNN | $0.665 \pm 0.03$ | 242K |
| FERNN-$V_1^T$ | $0.698 \pm 0.03$ | 242K |
| FERNN-$V_2^T$ | $\mathbf{0.716 \pm 0.04}$ | 242K |

Table 2: **FERNNs outperform non-equivariant models on sequence classification in the presence of strong motion**. Test accuracy (mean $\pm$ std) for models trained and tested on the Moving KTH dataset ($V_2^T$).

**Action Recognition Models.** In Figure 8 and Table 2, we compare the accuracy of FERNN models with non-flow-equivariant counterparts (G-RNN, G-RNN+, & 3D-CNN), each with comparable parameter counts. The G-RNN and FERNN-$V_*^T$ models are identical to those used on the Translating MNIST dataset with the addition of a 3-Layer CNN encoder (in place of the input convolution $\mathcal{U}$), and a class-readout module composed of an average-pooling over the spatial dimensions followed by a linear layer. The G-RNN+ model is an ablation of the FERNN-$V_2^T$ which has the same lifted hidden state and max-pooling over the lifted dimensions, but instead of applying the transformation $\psi_1(\nu)$ to each hidden index $\nu$, the G-RNN+ learns its own transformation through a small convolutional kernel (see §B). The inclusion of this model ensures that the benefits that we see from the FERNN are not simply due to having an increased hidden state size and an extra max-pooling non-linearity – instead they must be coming from the flow on the hidden state itself. The 3D-CNN is a 5-layer network composed of 3D convolutions that performs striding and pooling to eventually yield a 64-dimensional feature vector which is fed to a linear classification head.

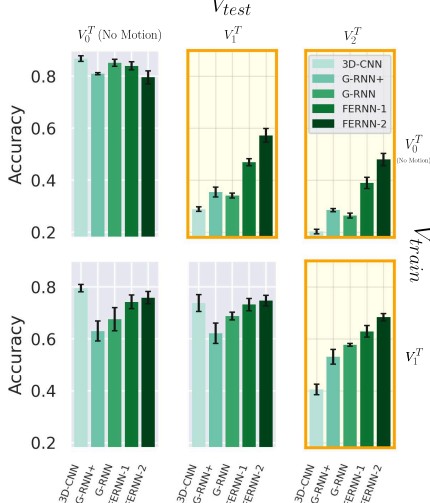

Figure 8: **FERNNs outperform non-equivariant models on action recognition undergoing out-of-distribution flows (orange)**. Test accuracy of models trained on KTH with no motion $V_0^T$ (top row) and with motion $V_1^T$ (bottom row); tested on $V_0^T$, $V_1^T$, $V_2^T$ (columns).

**Action Recognition Results** We see in Figure 8 that the FERNN models again outperform non-flow-equivariant models on generalization to classifying actions undergoing new flows at test time (plots highlighted in orange). We note that the drop in performance is likely due to the $V$ truncation discussed in §4 breaking exact equivariance and causing overfitting. In Table 2, we see that when trained and tested on strongly moving data, $V_{train} = V_{test} = V_2^T$, the FERNN significantly outperforms the non-flow-equivariant counterparts with no train-test disparity.

## 6 Discussion

In this work, we have introduced flow equivariance, a general notion of equivariance with respect to time-parameterized group transformations for sequence models, and provided an initial framework for constructing sequence models which satisfy this definition.

**Limitations & Future Work.** An immediately apparent limitation of FERNNs is the formal restriction to constant 'velocity' transformations. In Appendix §E.5 we show that in practice, FERNNs are still able to model variable velocities, and even inter-velocity interactions, through the use of inter-$V$ elements of the flow convolution kernels. A second significant limitation of the proposed FERNN construction is that it relies on the shift representation of the flow action on the $\nu$ dimension. This implies that the number of activations, compute, and memory must linearly increase with the size of $V$ (see §B.9), and additionally induces the truncation errors described in §4. Future work on an analog to Steerable CNNs [Cohen and Welling, 2016b] for flow equivariance would be highly valuable in alleviating this limitation. In summary, many of the FERNN's limitations are nearly identical to those of the original Group Equivariant Neural Networks – namely, scalability and discrete group equivariance. With time, however, researchers have developed methods to ameliorate these limitations, such as through steerability and gauge equivariance [Cohen et al., 2019]. In doing so, group equivariance has become practically relevant, including being used in highly popularized models [Jumper et al., 2021]. We therefore suggest that our paper may be seen as a first step in the introduction of a new theoretical foundation for dynamic equivariance in sequence models.

**Related Work.** While our definition of flow equivariance is novel, prior work has developed equivariance with respect to space-time transformations such as Lorentz transformations [Bogatskiy et al., 2020, Gong et al., 2022]; essentially processing the input as a 4D spacetime block such that the transformation is therein self-contained. The novelty of our work compared to this literature is the fact that our approach applies to causal sequence processing, and is defined for arbitrary sequence models rather than spatio-temporal convolutions only. A number of works consider equivariance in recurrent frameworks, [Azari and Erdoğmuş, 2022, Nguyen et al., 2023] but only consider static coordinate transformations, not those which change over time, as our flows (e.g. Figure 1). Closest to our work, 'Equivariant Observer Theory' [Mahony et al., 2022] has studied how to build hand-crafted models and filters which are indeed equivariant with respect to time-parameterized symmetries; however, this work lacks a notion of training familiar to the geometric deep learning community. Notably, several biologically inspired sequence models, including the Topographic VAE [Keller and Welling, 2022] and Neural Wave Machine [Keller and Welling, 2023], explicitly 'roll' latent representations so that a single-velocity flow is baked into the network and subsequently learned from data. Our formulation suggests that these heuristics may be special cases of Flow Equivariant RNNs and generalizes the principle to any set of continuous flows $V$. We refer readers to §D for extended discussion.

**Relation to Traveling Waves in Biological Neural Networks.** Spatiotemporal flows of neural activity, such as traveling waves, have been observed throughout the brain since the earliest recordings [Adrian and Matthews, 1934, Muller et al., 2018]. Related work in the domain of neuroscience has studied how such dynamics can be used to facilitate encoding visual motion [Jancke et al., 2004, Chemla et al., 2018, Townsend et al., 2017, Heitmann and Ermentrout, 2020, Benigno et al., 2023, May and Gjorgjieva, 2024], physical motion [Kaske and Bertschinger, 2005, Rubino et al., 2006], or motion through abstract spaces [Keller et al., 2024b]. Our work can be seen to formalize this previously observed connection between motion and flowing neural dynamics in the language of equivariant neural network theory. Specifically, the study of flow equivariance shows that autonomous traveling-wave-like dynamics (flows) are not only useful for encoding motion, but rather they are *necessary* for the accurate and stable representation of any stimulus undergoing a corresponding flow. In §3, we have shown that for any non-trivial recurrent neural network to process a sequence undergoing a flow transformation (such as visual motion) in a structured equivariant manner, the hidden state dynamics *must* have hidden-state dynamics that *autonomously* realize a homomorphic representation of the same flow, rather than being input-driven to emulate it. In addition to this theoretical argument, we highlight the relationship between lie-group structured flows (our $\psi_t(\nu)$) and traveling waves in the neuroscience literature. Specifically, it has been found that the lateral recurrent connections in cortex, along which spontaneous traveling waves in visual cortex are believed to propagate [Davis et al., 2024], preferentially connect neurons along smooth integral curves of lie group transformations with respect to individual neural selectivity [Ben-Shahar and Zucker, 2004, Seriès et al., 2002, Hoffman, 1989]. In other words, the connections which are believed to support latent flows are known to connect neurons with receptive fields which are precisely related by the representation of the flow on the input space. Altogether, we believe this combined evidence supports serious consideration of the hypothesis that some forms of observed spatiotemporal neural dynamics may be functioning as a form of time-parameterized symmetry equivariance, and we hope this encourages further empirical investigation in the future.

## Acknowledgments

We would like to thank Max Welling for continued guidance and advice, which ultimately led the author to the development of this work. We would also like to thank Maurice Weiler, John Vastola, and Gyu Heo for discussions which improved this work. We would like to thank members of the CRISP Lab at Harvard, and the Kempner Institute for the opportunity to present and improve early versions of the work. Finally, we would like to thank the Kempner Institute for funding, and for the opportunity to independently pursue the development of this work through the author's research fellowship. This work has been made possible in part by a gift from the Chan Zuckerberg Initiative Foundation to establish the Kempner Institute for the Study of Natural and Artificial Intelligence at Harvard University.

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

# Appendix

# A Extended Proofs

## A.1 Equivariance of Group-Lifting Convolution & Group-Convolution

We can verify that group convolutional layers given by Equation 3 are equivariant according to Equation 2. While these are well-known facts, we re-derive them here in our own notation for completeness and to lay the groundwork for our later proofs. First, for the lifting convolution, we wish to prove:

$$[(\hat{g} \cdot f) \hat{\star}_G \mathcal{W}^i](g) = \hat{g} \cdot [f \hat{\star}_G \mathcal{W}^i](g) \quad \forall g, \hat{g} \in G, \; f \in \mathcal{F}_K(X). \tag{15}$$

*Proof.* (Group-Lifting Conv. is Group Equivariant)

$$[(\hat{g} \cdot f) \hat{\star}_G \mathcal{W}^i](g) = \sum_{x \in \mathbb{Z}^2} \sum_{k=1}^{K} f_k(\hat{g}^{-1} \cdot x) \mathcal{W}_k^i(g^{-1} \cdot x) \quad \text{(by def. action, Eqn. 1)} \tag{16}$$

$$= \sum_{\hat{x} \in \mathbb{Z}^2} \sum_{k=1}^{K} f_k(\hat{x}) \mathcal{W}_k^i(g^{-1} \cdot (\hat{g} \cdot \hat{x})) \quad \text{(where } \hat{x} = \hat{g}^{-1} \cdot x\text{)} \tag{17}$$

$$= \sum_{\hat{x} \in \mathbb{Z}^2} \sum_{k=1}^{K} f_k(\hat{x}) \mathcal{W}_k^i((g^{-1} \cdot \hat{g}) \cdot \hat{x})) \quad \text{(by associativity)} \tag{18}$$

$$= \sum_{\hat{x} \in \mathbb{Z}^2} \sum_{k=1}^{K} f_k(\hat{x}) \mathcal{W}_k^i((\hat{g}^{-1} \cdot g)^{-1} \cdot \hat{x})) \quad \text{(by defn. inverse)} \tag{19}$$

$$= [f \hat{\star}_G \mathcal{W}^i](\hat{g}^{-1} \cdot g) = \hat{g} \cdot [f \hat{\star}_G \mathcal{W}^i](g) \quad \text{(by Eqn. 3 \& defn. action)} \tag{20}$$

$\square$

In line 17, we use the fact that because the group $G$ acts on $X$ by a bijection (formally, the input space representation of the group is $\pi_X : G \to \text{Aut}(X)$), the substitution $\hat{x} = g^{-1} \cdot x$ is just a relabeling of the index set, and hence the sum over $x \in X$ is the same as the sum over $\hat{x} \in X$.

Next for the group convolution, the proof is virtually identical. We wish to prove:

$$[(\hat{g} \cdot f) \star_G \mathcal{W}^i](g) = \hat{g} \cdot [f \star_G \mathcal{W}^i](g) \quad \forall g, \hat{g} \in G, \; f \in \mathcal{F}_K(X). \tag{21}$$

*Proof.* (Group-Conv. is Group Equivariant)

$$[(\hat{g} \cdot f) \star_G \mathcal{W}^i](g) = \sum_{h \in G} \sum_{k=1}^{K} f_k(\hat{g}^{-1} \cdot h) \mathcal{W}_k^i(g^{-1} \cdot h) \quad \text{(by def. action, Eqn. 1)} \tag{22}$$

$$= \sum_{\hat{h} \in G} \sum_{k=1}^{K} f_k(\hat{h}) \mathcal{W}_k^i(g^{-1} \cdot (\hat{g} \cdot \hat{h})) \quad \text{(where } \hat{h} = \hat{g}^{-1} \cdot h\text{)} \tag{23}$$

$$= \sum_{\hat{h} \in G} \sum_{k=1}^{K} f_k(\hat{h}) \mathcal{W}_k^i((g^{-1} \cdot \hat{g}) \cdot \hat{h})) \quad \text{(by associativity)} \tag{24}$$

$$= \sum_{\hat{h} \in G} \sum_{k=1}^{K} f_k(\hat{h}) \mathcal{W}_k^i((\hat{g}^{-1} \cdot g)^{-1} \cdot \hat{h})) \quad \text{(by defn. inverse)} \tag{25}$$

$$= [f \hat{\star}_G \mathcal{W}^i](\hat{g}^{-1} \cdot g) = \hat{g} \cdot [f \hat{\star}_G \mathcal{W}^i](g) \quad \text{(by Eqn. 3 \& defn. action)} \tag{26}$$

$\square$

Again, we use the fact that the group $G$ is closed under the group action, so the substitution $\hat{h} = \hat{g}^{-1} \cdot h$ is just a relabeling of the index set, and hence the sum over $h \in G$ is the same as the sum over $\hat{h} \in G$.

## A.2 Equivariance of Group Equivariant RNN

We wish to prove that a recurrent neural network built with group-equivariant convolutional layers (Equation 4) is equivariant according to Equation 2. If we write the hidden state as a function of the input signal $h_{t+1}[f_{\leq t}](g) \in \mathcal{F}_{K'}(Y)$, where $Y = G$, then we can write the the equivariance condition as:

$$h_{t+1}[\hat{g} \cdot f_{\leq t}](g) = \hat{g} \cdot h_{t+1}[f_{\leq t}](g) \quad \forall\, t \in \mathbb{Z}_+,\ f \in \mathcal{F}_K(X),\ \hat{g}, g \in G, \tag{27}$$

where $\hat{g} \cdot f_{\leq t} := \{\hat{g} \cdot f_i \mid i \in \mathbb{Z}_+ \leq t\}$, denotes the input signal generated by applying the same group element to each timestep. We prove this by induction for all $t \in \mathbb{Z}_+$:

*Proof.* (G-RNN is Group Equivariant)

We assume (i) $\hat{\star}_G$ and $\star_G$ are the group-lifting convolution and group convolution defined in Equation 3, (ii) $\sigma$ is a G-equivariant non-linearity (e.g. pointwise), and (iii) $g$ acts linearly on $h$-space. Since $h$ is defined by the lifting and recurrent convolutions, we see that $Y = G$. We further assume (iv) $h_0$ is initialized to be invariant to the group action, i.e. $\hat{g} \cdot h_0(g) = h_0(\hat{g}^{-1} \cdot g) = h_0(g)$, and (v) the input signal is zero before time zero, i.e. $f_{<0} = \mathbf{0}$.

Base Case: We can see that the base case is trivially true from the initial condition, since the initial condition is not a function of the input sequence, so:

$$h_0[\hat{g} \cdot f_{<0}](g) = h_0[f_{<0}](g) \quad \text{(since } h_0 \text{ is indep. of input } f_{<0}) \tag{28}$$

$$= h_0[f_{<0}](\hat{g}^{-1} \cdot g) \quad \text{(by initialization)} \tag{29}$$

$$= \hat{g} \cdot h_0[f_{<0}](g) \quad \text{(by defn. action)} \tag{30}$$

Inductive Step: Assuming $h_t[\hat{g} \cdot f_{<t}](g) = \hat{g} \cdot h_t[f_{<t}](g)$, we wish to prove this holds also for $t + 1$:

$$h_{t+1}[\hat{g} \cdot f_{\leq t}](g) = \sigma\big([h_t[\hat{g} \cdot f_{<t}] \star_G \mathcal{W}](g) + [[\hat{g} \cdot f_t]\hat{\star}_G \mathcal{U}](g)\big) \quad \text{(by defn. G-RNN)} \tag{31}$$

$$= \sigma\big([[\hat{g} \cdot h_t[f_{<t}]] \star_G \mathcal{W}](g) + [[\hat{g} \cdot f_t]\hat{\star}_G \mathcal{U}](g)\big) \quad \text{(by inductive hyp.)} \tag{32}$$

$$= \sigma\big(\hat{g} \cdot [h_t[f_{<t}] \star_G \mathcal{W}](g) + \hat{g} \cdot [f_t\hat{\star}_G \mathcal{U}](g)\big) \quad \text{(by equivar. of G-conv.)} \tag{33}$$

$$= \hat{g} \cdot \sigma\big([h_t[f_{<t}] \star_G \mathcal{W}](g) + [f_t\hat{\star}_G \mathcal{U}](g)\big) \quad \text{(by equivar. non-lin.)} \tag{34}$$

$$= \hat{g} \cdot h_{t+1}[f_{\leq t}](g) \quad \text{(by defn. G-RNN, Eqn 4)} \tag{35}$$

$$\square$$

We note that the step on the fourth line assumes that the group action is linear, in that it distributes over the addition operation between the previous hidden state and the input. Traditionally, group equivariant neural network architectures have been constructed to exhibit linear representations in the latent space, and therefore this is a natural assumption. However, if one allowed $g$ to act by non-linear maps (i.e. dropped the linear-representation assumption), then neither distributivity nor $\sigma$-equivariance would hold in general.

## A.3 Frame-wise Flow Equivariance

We wish to prove that any G-equivariant map $\phi : \mathcal{F}_K(X) \to \mathcal{F}_{K'}(Y)$, that is applied 'frame-wise' to a space-time function $f \in \mathcal{F}_K(X, \mathbb{Z})$ is also flow equivariant according to Definition 3.1. Specifically, let $\Phi[f] = [\phi(f_0), \phi(f_1) \dots \phi(f_T)]$ be a sequence model built by concatenating the output of a G-equivariant map $\phi$ applied to each timestep $t$ of the input signal $f_t$. Furthermore, let $\phi$ be equivariant to the individual group elements generated by vector fields $\nu \in V$, i.e. $\phi(\psi_t(\nu) \cdot f_t) = \psi_t(\nu) \cdot \phi(f_t)\ \forall t \in \mathbb{Z},\ \nu \in V, f_t \in \mathcal{F}_K(X)$. Then:

*Proof.* (Frame-wise Equivariant Maps are Flow Equivariant)

$$\Phi(\psi(\nu) \cdot f) = [\phi(\psi_0(\nu) \cdot f_0), \phi(\psi_1(\nu) \cdot f_1) \dots \phi(\psi_T(\nu) \cdot f_T)] \quad \text{(by defn. flow action)} \tag{36}$$

$$= [\psi_0(\nu) \cdot \phi(f_0), \psi_1(\nu) \cdot \phi(f_1) \dots \psi_T(\nu) \cdot \phi(f_T)] \quad \text{(by G-equivariance of } \phi) \tag{37}$$

$$= \psi(\nu) \cdot [\phi(f_0), \phi(f_1) \dots \phi(f_T)] \quad \text{(by defn. flow action, eqn. 5)} \tag{38}$$

$$= (\psi(\nu) \cdot \Phi(f)) \quad \text{(by defn. } \Phi) \tag{39}$$

$$\square$$

## A.4 Group Equivariant RNNs are not Flow Equivariant

In this subsection we prove Theorem 3.1, that the group-equivariant RNN as defined in Equation 4, with non-zero $\mathcal{W}$, is not flow equivariant according to Definition 3.1, except in the degenerate flow invariant case.

We will prove this in two parts: (i) First we will prove that a G-RNN with constant kernels is indeed flow invariant through induction (the degenerate case). Then (ii), we will proceed with a proof by contradiction to show that this is the only such flow equivariant G-RNN possible.

To begin, we recall that the degenerate flow invariant case is defined as a G-RNN where both $\mathcal{W}$ and $\mathcal{U}$ are constant over G. We will denote such kernels $\bar{\mathcal{U}}$ and $\bar{\mathcal{W}}$, such that:

$$\bar{\mathcal{W}}(g) = \bar{\mathcal{W}}(g'), \quad \forall g, g' \in G \quad \& \quad \bar{\mathcal{U}}(g \cdot x) = \bar{\mathcal{U}}(g' \cdot x) \quad \forall g, g' \in G, x \in X. \tag{40}$$

We wish to prove that such a network satisfies Definition 3.1, but more specifically that it is actually invariant to the flow action, meaning:

$$h_t[\psi(\nu) \cdot f_{<t}] = h_t[f_{<t}] \quad \forall f \in \mathcal{F}_K(X), \ \nu \in V, \ t \in \mathbb{Z}_+ \tag{41}$$

We can see that this is a special case of Definition 3.1 where the group action on the output space is trivial, i.e. given by the identity: $\psi_t(\nu) \cdot h_t[f_{<t}] = h_t[f_{<t}]$. We can prove this by induction as before. First we introduce the following lemma for convenience:

*Lemma* A.1. (Lifting convolution with $\bar{\mathcal{U}}$ is group invariant) The result of applying the lifting convolution $(\hat{\star}_G)$ defined in 3 with a constant kernel $\bar{\mathcal{U}}$ to a a signal $f \in \mathcal{F}_K(X)$ is invariant to flow element action on $f$, i.e.

$$[(\psi_t(\nu) \cdot f) \hat{\star}_G \bar{\mathcal{U}}^i](g) = [f \hat{\star}_G \bar{\mathcal{U}}^i](g) \quad \forall \nu \in V, \ f \in \mathcal{F}_K(X). \tag{42}$$

*Proof.* (Lemma A.1)

$$[(\psi_t(\nu) \cdot f) \hat{\star}_G \bar{\mathcal{U}}^i](g) = \sum_{x \in \mathbb{Z}^2} \sum_{k=1}^{K} f_k(\psi_t(\nu)^{-1} \cdot x) \bar{\mathcal{U}}_k^i(g^{-1} \cdot x) \quad \text{(by def. action, Eqn. 1)} \tag{43}$$

$$= \sum_{\hat{x} \in \mathbb{Z}^2} \sum_{k=1}^{K} f_k(\hat{x}) \bar{\mathcal{U}}_k^i(g^{-1} \cdot (\psi_t(\nu) \cdot \hat{x})) \quad \text{(where } \hat{x} = \psi(\nu)^{-1} \cdot x) \tag{44}$$

$$= \sum_{\hat{x} \in \mathbb{Z}^2} \sum_{k=1}^{K} f_k(\hat{x}) \bar{\mathcal{U}}_k^i((\psi_t(\nu)^{-1} \cdot g)^{-1} \cdot \hat{x})) \quad \text{(by associativity and inv.)}$$

$$\tag{45}$$

$$= \sum_{\hat{x} \in \mathbb{Z}^2} \sum_{k=1}^{K} f_k(\hat{x}) \bar{\mathcal{U}}_k^i(\hat{g}^{-1} \cdot \hat{x})) \quad \text{(by closure, } \hat{g} = \psi_t(\nu)^{-1} \cdot g \in G) \tag{46}$$

$$= \sum_{\hat{x} \in \mathbb{Z}^2} \sum_{k=1}^{K} f_k(\hat{x}) \bar{\mathcal{U}}_k^i(g^{-1} \cdot \hat{x})) \quad \text{(by defn } \bar{\mathcal{U}}(g^{-1} \cdot \hat{x}) = \bar{\mathcal{U}}(\hat{g}^{-1} \cdot \hat{x})) \tag{47}$$

$$= [f \hat{\star}_G \bar{\mathcal{U}}^i](g) \quad \text{(by Eqn. 3)} \tag{48}$$

$$\square$$

We have again used the fact that the group acts on $X$ by a bijection as in §A.1. We can then proceed with the proof by induction to prove that the G-RNN with constant kernels is flow invariant:

*Proof.* (G-RNN with Constant Kernels is Flow Invariant)

Let $h_t[f_{<t}]$ be defined as in Equation 4 with $\mathcal{U} = \bar{\mathcal{U}}$ and $\mathcal{W} = \bar{\mathcal{W}}$. We again assume (i) $\hat{\star}_G$ and $\star_G$ are the group-lifting convolution and group convolution defined in Equation 3, (ii) $\sigma$ is a G-equivariant non-linearity (e.g. pointwise), (iii) $g$ acts linearly on $h$-space, (iv) $h_0$ is initialized to be invariant to the group action of the individual flow elements, i.e. $\psi_t(\nu) \cdot h_0(g) = h_0(\psi_t(\nu)^{-1} \cdot g) = h_0(g)$, and (v) the input signal is zero before time zero, i.e. $f_{<0} = \mathbf{0}$.

Base Case: We can see that the base case is again trivially true from the initial condition, since the initial condition is not a function of the input sequence, so:

$$h_0[\psi(\nu) \cdot f_{<0}](g) = h_0[f_{<0}](g) \quad \text{(since } h_0 \text{ is indep. of input } f_{<0}) \tag{49}$$

Inductive Step: Assuming the inductive hypothesis that $h_t[\psi(\nu) \cdot f_{<t}] = h_t[f_{<t}]$, we wish to prove this holds for $t + 1$:

$$h_{t+1}[\psi(\nu) \cdot f_{\leq t}](g) = \sigma\big([h_t[\psi(\nu) \cdot f_{<t}] \star_G \bar{\mathcal{W}}](g) + [[\psi_t(\nu) \cdot f_t]\hat{\star}_G \bar{\mathcal{U}}](g)\big) \quad \text{(by defn. G-RNN)} \tag{50}$$

$$= \sigma\big([h_t[f_{<t}] \star_G \bar{\mathcal{W}}](g) + [[\psi_t(\nu) \cdot f_t]\hat{\star}_G \bar{\mathcal{U}}](g)\big) \quad \text{(by inductive hyp.)} \tag{51}$$

$$= \sigma\big([h_t[f_{<t}] \star_G \bar{\mathcal{W}}](g) + [f_t\hat{\star}_G \bar{\mathcal{U}}](g)\big) \quad \text{(by Lemma A.1)} \tag{52}$$

$$= h_{t+1}[f_{\leq t}](g) \quad \text{(by eqn. 4)} \tag{53}$$

$$\square$$

Finally then, we use a proof by contradiction to show that this degenerate flow invariant case is the only possible case for the G-RNN to be flow equivariant.

*Proof.* (Theorem 3.1, G-RNNs are not Generally Flow Equivariant)

Assert the converse, that the G-RNN defined in Equation 4 is flow equivariant, and that the kernels $\mathcal{W}$ and $\mathcal{U}$, are not constant, i.e. $\mathcal{W} \neq \bar{\mathcal{W}}$ & $\mathcal{U} \neq \bar{\mathcal{U}}$ (as defined in Equation 40). Then, it should be the case that:

$$h_{t+1}[\psi(\nu) \cdot f_{\leq t}](g) = (\psi(\nu) \cdot h[f_{\leq t}])_{t+1}(g) = h_{t+1}[f_{\leq t}](\psi_t(\nu)^{-1} \cdot g) \tag{54}$$

For $t = 0$ we have the following:

$$h_1[\psi(\nu) \cdot f_{\leq 0}](g) = \sigma\big([h_0[\psi(\nu) \cdot f_{<0}] \star_G \mathcal{W}](g) + [[\psi_0(\nu) \cdot f_0]\hat{\star}_G \mathcal{U}](g)\big) \tag{55}$$

$$= \sigma\big([h_0[f_{<0}] \star_G \mathcal{W}](g) + [f_0\hat{\star}_G \mathcal{U}](g)\big) \quad \text{(by defn. } h_0 \text{ & } \psi_0) \tag{56}$$

$$= h_1[f_{\leq 0}](g) \tag{57}$$

For the next step, we get:

$$h_2[\psi(\nu) \cdot f_{\leq 2}](g) = \sigma\big([h_1[\psi(\nu) \cdot f_{\leq 0}] \star_G \mathcal{W}](g) + [[\psi_1(\nu) \cdot f_1]\hat{\star}_G \mathcal{U}](g)\big) \tag{58}$$

$$= \sigma\big([h_1[f_{\leq 0}] \star_G \mathcal{W}](g) + \psi_1(\nu) \cdot [f_1\hat{\star}_G \mathcal{U}](g)\big) \quad \text{(by above & g-conv.)} \tag{59}$$

If we look at the action of the flow on the output space, according to flow equivariance, we should have:

$$(\psi(\nu) \cdot h[f_{\leq 2}])_2(g) = \sigma\big(\psi_1(\nu) \cdot [h_1[f_{\leq 0}] \star_G \mathcal{W}](g) + [[\psi_1(\nu) \cdot f_1]\hat{\star}_G \mathcal{U}](g)\big) \tag{60}$$

$$= \sigma\big(\psi_1(\nu) \cdot [h_1[f_{\leq 0}] \star_G \mathcal{W}](g) + \psi_1(\nu) \cdot [f_1\hat{\star}_G \mathcal{U}](g)\big) \quad \text{(by g-conv.)} \tag{61}$$

Setting $h_2[\psi(\nu) \cdot f_{\leq 2}](g) = (\psi(\nu) \cdot h[f_{\leq 2}])_2(g)$, as per our assumption, we see the input terms match exactly, but the $h_{t-1}$ terms imply the following equivalence:

$$[h_1[f_{\leq 0}] \star_G \mathcal{W}](g) = \psi_1(\nu) \cdot [h_1[f_{\leq 0}] \star_G \mathcal{W}](g) \tag{62}$$

$$= [h_1[f_{\leq 0}] \star_G \mathcal{W}](\psi_1(\nu)^{-1} \cdot g) \quad \text{(by action defn.)} \tag{63}$$

We see this is precisely the 'lagging' hidden state that we visualized in Figure 2. In order for this equality to hold for all $\nu \in V$, the output of the convolution must be constant along the flows generated by all $\nu$. We can see that this is only satisfied by $\mathcal{W} = \bar{\mathcal{W}}$, a contradiction.

$$\square$$

## A.5 Equivariance of Lifting Flow Convolution

In this section we verify that the flow-lifting convolution given by Equation 7 is equivariant to action of the individual flow elements, with the following representation of the action:

$$\psi_t(\nu) \cdot [f_t \hat{\star}_{V \times G} \mathcal{U}^i](\nu, g) = [(\psi_t(\nu) \cdot f_t) \hat{\star}_{V \times G} \mathcal{U}^i](\nu, g) := [f_t \hat{\star}_{V \times G} \mathcal{U}^i](\nu, \psi_t(\nu)^{-1} \cdot g) \quad (64)$$

Since the flow lifting convolution is a trivial lift, equivalent to the group-lifting convolution with an extra duplicated index, this proof is a trivial replication of the group-lifting convolution proof:

*Proof.* (Flow-Lifting Conv. is Flow Equivariant)

$$[(\psi_t(\nu) \cdot f_t) \hat{\star}_{V \times G} \mathcal{U}^i](\nu, g) = \sum_{x \in X} \sum_{k=1}^{K} f_k(\psi_t(\nu)^{-1} \cdot x) \mathcal{U}_k^i(g^{-1} \cdot x) \quad \text{(by defn. action)} \quad (65)$$

$$= \sum_{\hat{x} \in X} \sum_{k=1}^{K} f_k(\hat{x}) \mathcal{U}_k^i(g^{-1} \cdot (\psi_t(\nu) \cdot \hat{x})) \quad \text{(where } \hat{x} = \psi_t(\nu)^{-1} \cdot x) \tag{66}$$

$$= \sum_{\hat{x} \in X} \sum_{k=1}^{K} f_k(\hat{x}) \mathcal{U}_k^i((\psi_t(\nu)^{-1} \cdot g)^{-1} \cdot \hat{x})) \quad \text{(by associativity \& inv.)} \tag{67}$$

$$= [f_t \hat{\star}_{V \times G} \mathcal{U}^i](\nu, (\psi_t(\nu)^{-1} \cdot g)) \tag{68}$$

$\square$

## A.6 Equivariance of Flow Convolution

In this section, we prove that the flow convolution in Equation 9 is equivariant to the action of the individual flow elements yielding:

$$[(\psi_t(\nu) \cdot h) \star_{V \times G} \mathcal{W}^i](\nu, g) = \psi_t(\nu) \cdot [h \star_{V \times G} \mathcal{W}^i](\nu, g) \tag{69}$$

*Proof.* (Flow Conv. is Flow Equivariant)

$$[(\psi_t(\nu) \cdot h) \star_{V \times G} \mathcal{W}^i](\nu, g) = \sum_{\gamma \in V} \sum_{m \in G} \sum_{k=1}^{K'} h_k(\gamma, \psi_t(\nu)^{-1} \cdot m) \mathcal{W}_k^i(\gamma - \nu, g^{-1} \cdot m) \tag{70}$$

$$\text{(where } \hat{m} = \psi_t(\nu)^{-1} \cdot m) \quad = \sum_{\gamma \in V} \sum_{\hat{m} \in G} \sum_{k=1}^{K'} h_k(\gamma, \hat{m}) \mathcal{W}_k^i(\gamma - \nu, g^{-1} \cdot (\psi_t(\nu) \cdot \hat{m})) \tag{71}$$

$$\text{(by associativity \& inverse)} \quad = \sum_{\gamma \in V} \sum_{\hat{m} \in G} \sum_{k=1}^{K'} h_k(\gamma, \hat{m}) \mathcal{W}_k^i(\gamma - \nu, (\psi_t(\nu)^{-1} \cdot g)^{-1} \cdot \hat{m}) \tag{72}$$

$$\text{(by Eqn. 9)} \quad = [h \star_{V \times G} \mathcal{W}^i](\nu, \psi_t(\nu)^{-1} \cdot g) \tag{73}$$

$$\text{(by action)} \quad = \psi_t(\nu) \cdot [h \star_{V \times G} \mathcal{W}^i](\nu, g) \tag{74}$$

$\square$

## A.7 Equivariance of FERNN

In this section, we prove Theorem 4.1 by induction. We restate the theorem below:

*Theorem.* (FERNNs are flow equivariant) Let $h[f] \in \mathcal{F}_{K'}(Y, \mathbb{Z})$ be a FERNN as defined in Equations 7, 9, and 11, with hidden-state initialization invariant to the group action and constant in the flow dimension, i.e. $h_0(\nu, g) = h_0(\nu', g) \forall \nu', \nu \in V$ and $\psi_1(\nu) \cdot h_0(\nu, g) = h_0(\nu, g) \forall \nu \in V, g \in G$. Then, $h[f]$ is flow equivariant according to Definition 3.1 with the following representation of the action of the flow in the output space for $t \geq 1$:

$$(\psi(\hat{\nu}) \cdot h[f])_t(\nu, g) = h_t[f](\nu - \hat{\nu}, \psi_{t-1}(\hat{\nu})^{-1} \cdot g) \tag{75}$$

We note for the sake of completeness, that this then implies the follow equivariance relations:

$$h_t[\psi(\hat{\nu}) \cdot f](\nu, g) = h_t[f](\nu - \hat{\nu}, \psi_{t-1}(\hat{\nu})^{-1} \cdot g) = \psi_{t-1}(\hat{\nu}) \cdot h_t[f](\nu - \hat{\nu}, g) \tag{76}$$

*Proof.* (Theorem 4.1, FERNNs are flow equivariant)

Identical to the G-RNN, we assume that $\sigma$ is a G-equivariant non-linearity, $g$ acts linearly on $h$-space, $h_0(\nu, g)$ is defined constant as above, and the input signal is zero before time zero, i.e. $f_{<0} = \mathbf{0}$.

Base Case: The base case is trivially true from the initial condition:

$$h_0[\psi(\hat{\nu}) \cdot f_{<0}](\nu, g) = h_0[f_{<0}](\nu, g) \quad \text{(by initial cond.)} \tag{77}$$

$$= h_0[f_{<0}](\nu - \hat{\nu}, \psi_{t-1}(\hat{\nu})^{-1} \cdot g) \quad \text{(by constant init.)} \tag{78}$$

Inductive Step: Assuming $h_t[\psi(\hat{\nu}) \cdot f](\nu, g) = h_t[f](\nu - \hat{\nu}, \psi_{t-1}(\hat{\nu})^{-1} \cdot g) \, \forall \, \nu \in V, g \in G$, for some $t \geq 0$, we wish to prove this also holds for $t + 1$:

Using the FERNN recurrence (Eqn. 11) on the transformed input, and letting $H = V \times G$, we get:

$$h_{t+1}[\psi(\hat{\nu}) \cdot f](\nu, g) = \sigma\Big(\psi_1(\nu) \cdot [h_t[\psi(\hat{\nu}) \cdot f_{<t}] \star_H W](\nu, g) + [(\psi_t(\hat{\nu}) \cdot f_t) \hat{\star}_H U](\nu, g)\Big) \tag{79}$$

$$\text{(by inductive hyp.)} \quad = \sigma\Big(\psi_1(\nu) \cdot [(\psi_{t-1}(\hat{\nu}) \cdot h_t[f_{<t}]) \star_H W](\nu - \hat{\nu}, g) + [(\psi_t(\hat{\nu}) \cdot f_t) \hat{\star}_H U](\nu, g)\Big) \tag{80}$$

$$\text{(by trivial inp. lift)} \quad = \sigma\Big(\psi_1(\nu) \cdot [(\psi_{t-1}(\hat{\nu}) \cdot h_t[f_{<t}]) \star_H W](\nu - \hat{\nu}, g) + [(\psi_t(\hat{\nu}) \cdot f_t) \hat{\star}_H U](\nu - \hat{\nu}, g)\Big) \tag{81}$$

$$\text{(by equiv. flow-conv)} \quad = \sigma\Big(\psi_1(\nu) \cdot \psi_{t-1}(\hat{\nu}) \cdot [h_t[f_{<t}] \star_H W](\nu - \hat{\nu}, g) + \psi_t(\hat{\nu}) \cdot [f_t \hat{\star}_H U](\nu - \hat{\nu}, g)\Big) \tag{82}$$

$$\text{(by flow properties)} \quad = \sigma\Big(\psi_t(\hat{\nu}) \cdot \psi_1(\nu - \hat{\nu})[h_t[f_{<t}] \star_H W](\nu - \hat{\nu}, g) + \psi_t(\hat{\nu}) \cdot [f_t \hat{\star}_H U](\nu - \hat{\nu}, g)\Big) \tag{83}$$

$$\text{(by eqiv. non-lin.)} \quad = \psi_t(\hat{\nu}) \cdot \sigma\Big(\psi_1(\nu - \hat{\nu})[h_t[f_{<t}] \star_H W](\nu - \hat{\nu}, g) + [f_t \hat{\star}_H U](\nu - \hat{\nu}, g)\Big) \tag{84}$$

$$\text{(by FERNN Eqn. 11)} \quad = \psi_t(\hat{\nu}) \cdot h_{t+1}[f](\nu - \hat{\nu}, g) \tag{85}$$

$$= h_{t+1}[f]\big(\nu - \hat{\nu}, \, \psi_t(\hat{\nu})^{-1} \cdot g\big), \tag{86}$$

Thus, assuming the inductive hypothesis for time $t$ implies the desired relation at time $t + 1$; together with the base case this completes the induction and proves Theorem 4.1. $\qquad\square$

Similar to the main text, we additionally provide a visual example which demonstrates that the counterexample to equivariance of the G-RNN now no longer holds for the FERNN. As we see in Figure 9, the additional $V$ dimension results in the moving input being picked up as if it were stationary in the corresponding channel. The $V$ channels can then be seen to permute as a result of the action of the flow in the latent space.

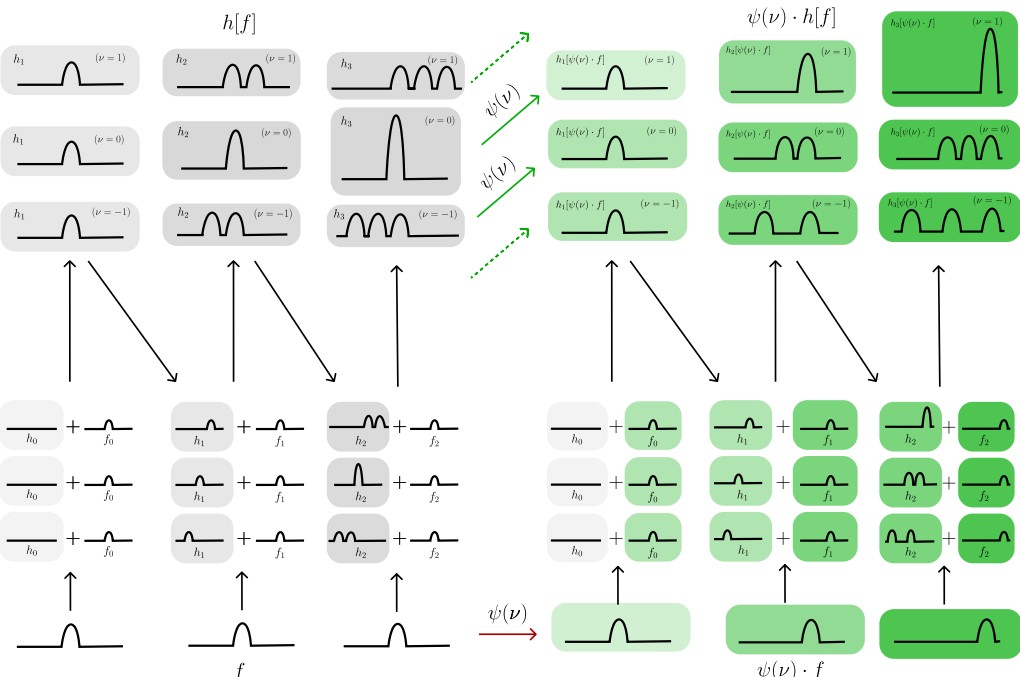

Figure 9: A visualization of the same counter example of Figure 2 from the main text, but for the Flow Equivariant RNN. We see that the model is built with a set of extra $V$ channels, depicted as the three separate rows of hidden states, where each row flows independently per timestep according to its $\nu$ parameter (e.g. the bottom row flows with velocity $-1$, to the left, while the top row flows with velocity $1$ to the right). When the model is then processing an input with a corresponding flow symmetry (for example a motion of velocity $1$ to the right), the corresponding flowing channel of the hidden state processes this input in the same reference frame, as if it were stationary (the top row in this example). The remaining rows then process the input with a difference corresponding to the difference in velocity between the hidden state channel and the input velocity. We see that this results in the $\nu - \hat{\nu}$ channel permutation in the latent space (the vertical shift in the $V$ dimension by 1 row).

# B    Experiment Details

In this section we describe the datasets, models, training and evaluation procedures used in §5 of the main text. We additionally include samples from each dataset for visualization. The full code to reproduce these experiments is available at: `https://github.com/akandykeller/FERNN`

## B.1    Flowing MNIST: Dataset Creation

As described in the main text, we construct sequences from the Flowing MNIST dataset by applying a flow generators $\nu$ randomly picked from an admissible set $V_{train}$, $V_{val}$, & $V_{test}$ to samples from the corresponding train / validation / test split of the original MNIST dataset [LeCun et al., 1998]. The training sequences are always composed of $T = 20$ time-steps, and identically for the test sequences (except in the length-generalization experiments where the test sequence length is increased to 40 time-steps). We similarly generally set $V_{train} = V_{val} = V_{test}$, except in the velocity generalization experiments, where we specify the training and test flows explicitly. Note that the set $V$ which defines the set of flows to which the FERNNs are equivariant can be anything, and must not match the $V_{train}$ of the dataset. Explicitly, denoting the value of the $i$th MNIST training image at pixel coordinate $(x, y)$ as $m_{train}^{(i)}((x, y))$, a sample from the training set can be written as:

$$f_t^{(i)}((x, y)) = \psi_t(\nu^1) \cdot m_{train}^{(j)}((x, y)) + \psi_t(\nu^2) \cdot m_{train}^{(k)}((x, y)) \quad \forall\, (x, y) \in \mathbb{Z}^2,\ t \in \mathbb{Z}_+ \leq T \quad (87)$$

where $\nu^1, \nu^2 \sim V_{train}$ and $j, k$ and random indices sampled with replacement from the MNIST dataset. We note that above, as in the main paper, we have written the signals and the kernels on the infinite domain $\mathbb{Z}^2$. In practice, however, both are only non-zero on a small portion of this domain (i.e. between $(0, 0)$ & $(28, 28)$ for MNIST digits). We do this to simplify the analysis, analogous to prior work [Cohen and Welling, 2016a].

**Translating MNIST.**    For the 2D-translation variant of Flowing MNIST, we consider the group

$$G = T(2) = (\mathbb{Z}^2, +), \qquad \text{with binary operation} \qquad (x, y) + (x', y') = (x + x', y + y'). \quad (88)$$

We use flow generators which are elements of the Lie algebra of this group, $\nu \in \mathfrak{t}(2)$, which we similarly denote as vectors in $\mathbb{Z}^2$, with the Lie bracket $[\gamma, \nu] = 0 \quad \forall\, \gamma, \nu \in \mathfrak{t}(2)$, meaning the translations are commutative. We note that in a matrix representation, we embed the elements of $T(2)$ into the affine group with homogeneous $3 \times 3$ matrices:

$$(x, y) \longmapsto \begin{pmatrix} 1 & 0 & x \\ 0 & 1 & y \\ 0 & 0 & 1 \end{pmatrix}, \qquad (89)$$

where the corresponding Lie algebra elements are

$$X(\nu) = \begin{pmatrix} 0 & 0 & \nu_1 \\ 0 & 0 & \nu_2 \\ 0 & 0 & 0 \end{pmatrix}, \qquad \nu = (\nu_1, \nu_2) \in \mathbb{Z}^2. \qquad (90)$$

The exponential map is then simply: $\exp(X(\nu)) = I + X(\nu)$. We see that the flow is then given as: $\psi_t(\nu) = \exp(tX(\nu)) = I + tX(\nu)$, and the action of the flow on a given pixel coordinate $g = (x, y)$ is given as:

$$\psi_t(\nu) \cdot g = t \begin{pmatrix} 1 & 0 & \nu_1 \\ 0 & 1 & \nu_2 \\ 0 & 0 & 1 \end{pmatrix} \begin{pmatrix} 1 & 0 & x \\ 0 & 1 & y \\ 0 & 0 & 1 \end{pmatrix} = \begin{pmatrix} 1 & 0 & x + t\nu_1 \\ 0 & 1 & y + t\nu_2 \\ 0 & 0 & 1 \end{pmatrix}, \qquad (91)$$

i.e. a shift of the pixel coordinates by velocity $\nu$ for $t$ time-steps. In practice, order to be able to generate long sequences without excessively large images, we perform all translations with cyclic boundary conditions on our input and hidden states, i.e. $\psi_t(\nu) \cdot (x, y) = \big( (x + t\nu_1) \mod W,\ (y + t\nu_2) \mod H \big)$, for image size $(H, W)$. Since all of our convolutions are also performed with cyclic boundary conditions, this does not impact performance.

In our experiments, to define the sets of generators that we are interested in, we always use the integer lattice up to velocities $N \frac{pixels}{step}$. We denote these sets with the notation:

$$V_N^T := \{\nu \in \mathbb{Z}^2 \mid ||\nu||_\infty \leq N\}. \qquad (92)$$

For example $V_2^T$ is the set of all 2D translation vectors with maximal velocity component $\pm 2$ in either dimension, i.e. $V_2^T = \{(-2, -2), (-2, -1), (-2, 0), \dots (2, 2)\}$.

**Rotating MNIST.** For the planar rotation variant of Flowing MNIST, we consider the group

$$G = SO(2) = (\mathbb{R}, +), \qquad \text{with binary operation} \qquad \theta + \theta' = (\theta + \theta') \mod 2\pi. \quad (93)$$

Elements of the Lie algebra $\mathfrak{so}(2)$ are one-parameter generators of in-plane rotations, each of the form $\nu = \omega J$, where $J = \begin{pmatrix} 0 & -1 \\ 1 & 0 \end{pmatrix}$ and $\omega \in \mathbb{Z}$ denotes the (integer-scaled) angular velocity. Because $\mathfrak{so}(2)$ is abelian, the Lie bracket $[\gamma, \nu] = 0 \;\; \forall\, \gamma, \nu \in \mathfrak{so}(2)$.

We embed $SO(2)$ into the affine group with $2 \times 2$ matrices:

$$\theta \longmapsto \begin{pmatrix} \cos\theta & -\sin\theta \\ \sin\theta & \cos\theta \end{pmatrix}, \quad (94)$$

whose corresponding Lie-algebra elements are

$$X(\nu) = \begin{pmatrix} 0 & -\omega \\ \omega & 0 \end{pmatrix}, \qquad \nu = \omega J, \;\; \omega \in \mathbb{Z}. \quad (95)$$

Since $X(\nu)^2 = -\omega^2 I_{2\times 2}$, the exponential map is the usual matrix exponential for planar rotations:

$$\exp\big(X(\nu)\big) = I + \sin(\omega)\, J + \big(1 - \cos(\omega)\big) J^2.$$

Hence the flow generated by $\nu$ is $\psi_t(\nu) = \exp\big(t\, X(\nu)\big)$, and its action on a pixel coordinate $g = (x, y)$ (in homogeneous form) is

$$\psi_t(\nu) \cdot g = \begin{pmatrix} \cos(t\omega) & -\sin(t\omega) \\ \sin(t\omega) & \cos(t\omega) \end{pmatrix} \begin{pmatrix} x \\ y \end{pmatrix} = \begin{pmatrix} x\cos(t\omega) - y\sin(t\omega) \\ x\sin(t\omega) + y\cos(t\omega) \end{pmatrix}, \quad (96)$$

i.e. a rotation about the image center by angle $t\,\omega$.

Following our experimental protocol, we discretize the angular velocity at $\Delta\theta = 10°$ intervals and collect the set of generators

$$V_N^R := \big\{\, \nu = k\,\Delta\theta\, J \,\big|\, k \in \mathbb{Z}, \; |k| \le N \big\}. \quad (97)$$

For instance, $V_4^R$ would consist of the set of angular velocities $\{-40°, -30°, -20°, -10°, 0°, 10°, 20°, 30°, 40°\}\frac{\deg}{\text{step}}$. In practice, to implement the spatial rotation we use the Pytorch function F.grid_sample with zero-padding and bilinear interpolation. We additionally zero-pad all images with 6-pixels on each side (resulting in images of size $(40 \times 40)$) to allow for the rotation to fit within the full image frame.

## B.2 Flowing MNIST: Models

For the Flowing MNIST datasets we compare three types of models: standard group-equivariant RNNs (G-RNNs) as defined in Equation 4, FERNNs with equivariance to a subset of the training flows (i.e. $V_{model} = V_1^T$ and $V_{train} = V_2^T$), and FERNNs with full equivariance to the training flows ($V_{model} = V_{train}$). For all models on each dataset we use the same model architecture, and since there are no extra parameters introduced by the FERNN model, all models have the same number of trainable parameters.

**Translating MNIST G-RNN.** For the translation group, the corresponding group-convolution is the standard 2D convolution. We therefore build a G-RNN exactly as written in Equation 4 with regular convolutional layers in place of the group-convolution. We use kernel sizes of $3 \times 3$ for both $\mathcal{U}$ and $\mathcal{W}$ with no bias terms, strides of 1 and circular padding of 1, resulting in a hidden state with spatial dimensions equal to the input spatial dimensions: $28 \times 28$. We use 128 output channels for our convolutional 'encoder' $\mathcal{U}$, and similarly 128 input and output channels for our recurrent kernel $\mathcal{W}$. This results in a hidden state $h \in \mathbb{R}^{128 \times 28 \times 28}$. We use a ReLU activation function: $\sigma(h) = \max(0, h)$. We initialize all hidden states to zero: $h_0 = \mathbf{0}$, satisfying our equivariance proof assumptions. At each timestep, we decode the updated hidden state to predict the next input through a 4-layer CNN decoder: $g_\theta(h_{t+1}) = \hat{f}_{t+1}$. The CNN decoder is composed of three primary convolutional layers each with 128 input and output channels, kernel size $3 \times 3$, stride 1, and circular padding 1, followed by ReLU activations. A final identical convolutional layer, but with 1 output channel, is used to predict $\hat{f}_{t+1}$.

**Translating MNIST FERNNs.** For the FERNNs, we use the exact same RNN and decoder architecture, with the only difference being that we extend the hidden state with an extra $V$ dimension for each of the flows in $V_{model}$: $h(\nu, g)$. Explicitly then, through the trivial lift, the input is copied identically to each of these $\nu$ channels, and the flow convolution on the hidden state also operates identically on each $\nu$ channel. As stated in the main text, we define the flow convolution to not mix the $V$ channels at all, explicitly: $\mathcal{W}_k^i(\nu, g) = \delta_{\nu=e}\mathcal{W}_k^i(g)$, thus giving us constant parameter count. Our lifted hidden state is thus $h \in \mathbb{R}^{|V_{model}| \times 128 \times 28 \times 28}$. The flow action $\psi_1(\nu)\cdot$ in the recurrence is implemented practically as a Roll of the hidden state tensor by $(\nu_1, \nu_2)$ steps along the $(x, y)$ spatial dimensions. To achieve invariance of the reconstruction to the input flows, and thereby achieve the generalization we report, we max-pool over the $V$ dimensions before decoding. Explicitly, the output of the model for each time step is computed as: $g_\theta(\max_\nu h_{t+1}(\nu, g)) = \hat{f}_{t+1}$.

**Rotating MNIST G-RNN.** For the rotating MNIST models, we use a nearly identical setup to the translation experiments, with the only difference being that we use $SE(2)$ group convolutions in place of the standard convolutional layers to achieve the necessary rotation equivariance. In practice, we use the *escnn* library to implement the $SE(2)$ convolutions [Cesa et al., 2022]. We discretize the rotation group into $\Delta\theta = 10^o$ rotations, yielding a cyclic group with 36 elements: $C_{36}$. We lift the input to this space and assert a regular representation of the group action on the output. Due to the increased dimensionality of the hidden state from the lift to the discrete rotation group, we decrease the number of hidden state channels to 32 due to hardware limitations. This then yields a hidden state: $h \in \mathbb{R}^{36 \times 32 \times 40 \times 40}$.

**Rotating MNIST FERNNs.** For the FERNN models, we follow the same procedure as for the translating MNIST FERNNs, except with the action of the group $\psi_1(\nu)\cdot$ in the recurrence now taking the form of the regular representation of rotation in the $SE(2)$ equivariant CNN output space. Explicitly, this means that in addition to rotating the inputs, we also permute along the lifted rotation channel for each angular velocity $\nu$. Again this yields a hidden state of size: $h \in \mathbb{R}^{|V_{model}| \times 36 \times 32 \times 40 \times 40}$.

### B.3 Flowing MNIST: Next-Step Prediction Training & Evaluation

For the in-distribution next step prediction experiments, (Table 1 and Figure 3) of the main text, we use $V_{train} = V_{val} = V_{test} = V_2^T$ for the translating MNIST experiments, and $V_{train} = V_{val} = V_{test} = V_4^R$ for the rotating MNIST experiments. We set the training sequence length to 20 steps, providing the models with 10 time-steps as input, and computing the next-step prediction reconstruction loss (MSE) of the model output on the remaining 10 time-steps. Explicitly:

$$\mathcal{L} = \frac{1}{10} \sum_{t=11}^{20} ||f_t - g_\theta(h_t)||_2^2 \tag{98}$$

All models are trained for 50 epochs, with a learning rate of $1 \times 10^{-4}$ using the Adam optimizer Kingma and Ba [2017]. For translation flows, we use a batch size of 128, and clip gradient magnitudes at 1 in all models for additional stability. For rotation flows we use a batch size of 32 due to memory constraints, and find gradient clipping not necessary. For evaluation, we save the model with the best performance on the validation set (over epochs), and report its corresponding performance on the held-out test set. For each model we train with 5 random initializations (5 seeds) and report the mean and standard deviation of the test set performance from the best saved models.

**Length Generalization.** For the out-of-distribution length generalization experiments (Figure 4) we again use $V_{train} = V_{val} = V_{test} = V_2^T$ for the translating MNIST experiments, and $V_{train} = V_{val} = V_{test} = V_4^R$ for the rotating MNIST experiments. Where the length of training and validation sequences is again set to $T = 20$ as before. At test time, we increase the length of the sequences to $T = 70$, but continue to only feed the models the first 10 time-steps as input. In Figure 4 we show the loss of the model for each of these remaining 60 time steps ahead.

**Velocity Generalization.** For the out-of-distribution velocity generalization experiments (Figure 5), we leave the train and test sequence length at 20, but we instead set $V_{train} = V_{val} \subseteq V_{test}$. Explicitly, in Figure 5, for rotating MNIST, we set $V_{train} = V_{val} = V_1^R$ and $V_{test} = V_5^R$. For translating

MNIST, we set $V_{train} = V_{val} = V_1^T$ and $V_{test} = V_2^T$. This tests the ability of models to generalize to new flows not seen during training, and we see that the FERNNs perform significantly better on this test set when they are made equivariant to these velocities.

## B.4 Moving KTH: Datasets

To test the benefits of flow equivariance on a sequence classification task with real image sequences, we opted to use the KTH action recognition dataset [Schuldt et al., 2004], obtained from `http://www.csc.kth.se/cvap/actions/`. The dataset is composed of 2391 videos of 25 people performing 6 different actions: running, jogging, walking, boxing, hand clapping, and hand waving. The original videos are provided at a resolution of $160 \times 120$ with 25 frames per second, and an average clip length of 4 seconds. The training, validation, and test sets are constructed from this dataset by taking the videos from the first 16 people as training, the next 4 people as validation, and the last 5 people as test. We split the videos into clips of 32-frames each, downsample to a spatial resolution of $32 \times 32$, and subsample them by half in time, yielding a final set of clips which are 16 steps long.

Since the videos from this dataset are taken entirely from a stationary camera viewpoint, there are no global flows of the input space which our model might benefit from. We call this original dataset KTH with $V_0^T$ (no motion). To test the benefits of flow equivariance in an action recognition setting with a moving viewpoint, we construct two additional variants of the KTH dataset, augmented by translation flows from the sets $V_1^T$ and $V_2^T$. These sets are identical to those described in the Flowing MNIST examples: translation with circular boundary conditions. Again, since we use convolution with circular boundary conditions in all our models, this does not impact model performance.

## B.5 Moving KTH: Models

We compare five models on the KTH dataset: a 3D-CNN, a G-RNN, two FERNN variants, and an ablation of the FERNN (denoted G-RNN+). We describe these in detail below:

**Moving KTH 3D-CNN.** The spatio-temporal 3D-CNN baseline is built as a sequence of five 3D convolution layers, interleaved with 3D batchnorm and ReLU activations. Each layer has kernels of shape $(3 \times 3 \times 3)$, no bias, and padding 1. The first layer has 16 output channels, a temporal stride of 2, and a spatial stride of 1. Layer 2 has 32 output channels, temporal stride of 1, and spatial stride of 2. Layer 3 has 32 output channels, temporal stride of 1, and spatial stride of 1. Layer 4 has 64 output channels, temporal stride of 1, and spatial stride of 2. Layer 5 has 64 output channels, temporal stride of 1, and spatial stride of 1. This final layer is followed by a global average pooling over the remaining $(8 \times 8 \times 8)$ space-time feature map dimensions, yeilding a single vector of dimensionality 64 which is passed through a linear layer to predict the logits for the 6 classes.

**Moving KTH G-RNN.** For the baseline G-RNN, each grayscale input frame $f_t \in \mathbb{R}^{1 \times 32 \times 32}$ is passed through a three-layer convolutional encoder that preserves spatial resolution:

$$\text{Conv}_{5 \times 5}^{1 \to 32} \xrightarrow{\text{BN+ReLU}} \text{Conv}_{3 \times 3}^{32 \to 64} \xrightarrow{\text{BN+ReLU}} \text{Conv}_{3 \times 3}^{64 \to 128} \xrightarrow{\text{BN+ReLU}},$$

all with stride 1 and circular padding chosen so that the hidden state has the same spatial dimensions as the input ($32 \times 32$). The output defines the encoder feature map $\mathcal{U} * f_t$. The hidden state $h_t \in \mathbb{R}^{128 \times 32 \times 32}$ is updated with a single recurrent layer using a circularly padded $3 \times 3$ convolution where $\mathcal{W} \in \mathbb{R}^{128 \times 128 \times 3 \times 3}$ contains no bias terms. The recurrent convolution is also followed by a batch-norm and ReLU non-linearity for added expressivity. The non-linearity $\sigma$ is $\tanh$. Initial states are zero, $h_0 = \mathbf{0}$, satisfying the assumptions of our equivariance proof. At the final timestep of the sequence ($t = 16$) we take the hidden state of RNN, perform global average pooling to a $1 \times 1$ spatial dimensions, and feed the resulting 128-dimensional vector through a fully-connected layer:

$$g_\theta(h_T) = \text{FC}_{128 \to 6}\big(\text{SpatialAvgPool}(h_T)\big) \in \mathbb{R}^6,$$

producing logits for the six KTH action classes.

**Moving KTH FERNNs.** For the FERNNs, we use the exact same architecture, but use the trivial lift to lift the corresponding sets of flows $V_1^T$ for FERNN-$V_1^T$ and $V_2^T$ for FERNN-$V_2^T$ (denoted FERNN-1 and FERNN-2 in Figure 8). Concretely, for every translation velocity $\nu \in V_{\text{model}}$ we allocate an additional velocity channel, so that the hidden state becomes $h_t(\nu, g) \in \mathbb{R}^{|V_{\text{model}}| \times 128 \times 32 \times 32}$,

where $g = (x, y)$ indexes spatial position. Input frames are trivially lifted by copying the same encoder features into each $\nu$-channel. Following Equation 11, the flow action in the recurrence is implemented again as a Roll of the spatial dimensions of the hidden tensor by $(\nu_x, \nu_y)$ pixels, and the flow convolution uses weight sharing across velocities, $\mathcal{W}_k^i(\nu, g) = \delta_{\nu=e} \mathcal{W}_k^i(g)$, so the total number of trainable parameters is identical to the G-RNN. As in the Flowing-MNIST experiments, we consider two settings: (i) *partial* equivariance with $V_{\text{model}} = V_1^T \subset V_{\text{train}} = V_2^T$ and (ii) *full* equivariance where $V_{\text{model}} = V_{\text{train}}$. To achieve flow-invariant action classification we take the maximum over the velocity dimension after the final FERNN layer, $\max_\nu h_t(\nu, g)$, followed by the same global-average-pooling and linear classifier used for the G-RNN. This design ensures that any translation-induced shifts present at test time are pooled over, yielding the generalization results reported in Figure 8.

**Moving KTH G-RNN+.** To ensure that the observed performance improvement of the FERNN was not simply due to the increased number of hidden state activations and the associated max-pooling, but instead could be attributed the precise flow equivariant form of the recurrence introduced in Equation 11, we built a third baseline which is as close as possible to the best performing FERNN (FERNN-$V_2^T$), while removing precise flow equivariance. Specifically, while keeping all other architectural components of the FERNN-$V_2^T$ identical, we replaced the single-step action of the flow in the recurrence ($\psi_1(\nu)\cdot$) with convolution by a separate learned $5 \times 5$ convolutional kernel for each $\nu$ channel (randomly inititalized). Since the action of $\psi_1(\nu)$ is a simple translation (a local linear operation), this can indeed be represented by such a kernel. However, as we see in practice (Figure 8), the model fails to learn such kernels and instead overfits to the training data distribution.

## B.6 Moving KTH: Action Recognition Training & Evaluation

Models are trained to minimize the cross entropy loss between the predicted class and the ground truth label using the Adam optimizer. Due to the small dataset size, all models are trained for 500 epochs, with a batch size of 32. We search over learning rates in the set $\{3 \times 10^{-3}, 1 \times 10^{-3}, 3 \times 10^{-4}, 1 \times 10^{-4}\}$, for each model, running three random initialization seeds for each. For each random seed of each hyper-parameter setting, we store the model with the best validation loss. We then pick the best performing model based on the mean value of the best validation loss across all three seeds. We then report in Table 2 the mean test loss of the models (3 seeds) saved at the best validation loss epoch for the best identified learning rate. We generally find the lower learning rates ($3 \times 10^{-4}$) work better for the RNN models, while the higher learning rates ($1 \times 10^{-3}$) work better for the CNNs.

## B.7 Moving KTH Samples

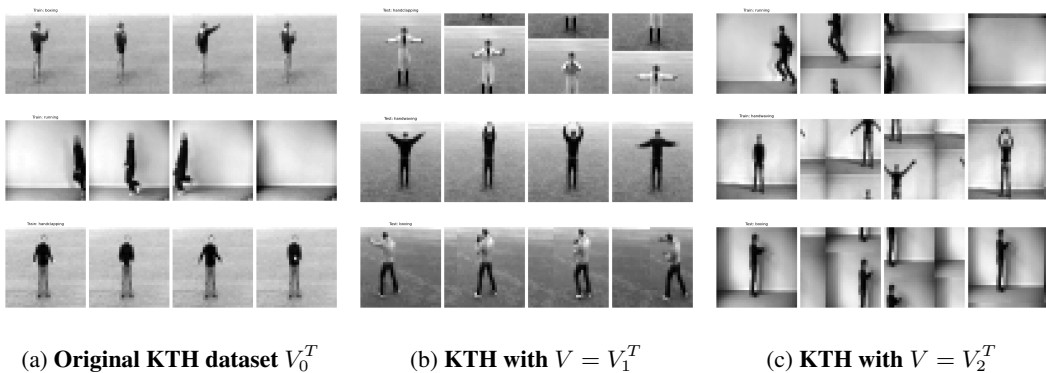

(a) **Original KTH dataset** $V_0^T$       (b) **KTH with** $V = V_1^T$       (c) **KTH with** $V = V_2^T$

Figure 10: Samples of the original KTH dataset and its two motion-augmented variants.

## B.8 Compute

All experiments in this paper were performed on a private cluster containing a mixture of NVIDIA A100 and H100 GPUs, each having 40GB and 80GB of VRAM respectively. No parallelization of

individual models across GPUs was required, i.e. most models and training paradigms were able to fit on a single A100 GPU, with the larger models on a single H100 GPU. The cluster nodes allocated up to 24 CPU cores and 375GB of RAM per job, although only a small fraction of this was required for training and evaluation. The majority of models were able to train fully in less than 24 hours. For example, the FERNN-$V_2^T$ models on KTH trained in 7 hours, and the FERNN-$V_2^T$ models on MNIST trained in 15 hours, with all other models training faster. The significant exception to this were the FERNN-$V_4^R$ models on Rotating MNIST which took roughly 67 hours to complete 50 epochs (although they converged much more quickly than this, see Figure 3, we ran them to the same number of epochs as the G-RNN for consistency). The reason for this increased computational time was an inefficient implementation of the rotation operation and our custom recurrence, which could both be accelerated in future work. Specifically, we used a naive vanilla Pytorch implementation of our custom FERNN recurrence (Equation 11) using 'for loops', which dramatically slowed down training for all models. In future work, implementation of the model with a scan operation in JAX, or a custom CUDA kernel would dramatically improve runtime performance. Overall, we estimate the computational requirements necessary to develop the models and run all experiments for this paper totaled approximately 30 days of H100 compute time.

### B.9 Runtime and Memory Usage vs. $|V|$

The flow convolution in Equation 9 performs a convolution over standard group elements ($g \in G$, which can be thought of as spatial positions for the simple translation case), and also over flow generators ($\nu \in V$, which can be thought of as movement velocities). This can be implemented efficiently as a 3D convolution for the case of 2D images, and a 1D generator group. For sets with higher dimensional generators, one can use N-D convolutions, which are also implemented efficiently in frameworks like Jax.

We predict the computational complexity of the model and the memory usage should scale linearly with the size of the set of generators $V$ (which we denote $|V|$). In the figure below we validate this by plotting the memory requirements and runtime per epoch as a function of the size of this set. Note the G-RNN is equivalent to a FERNN with $|V| = 1$, i.e. $V = \{\mathbf{0}\}$.

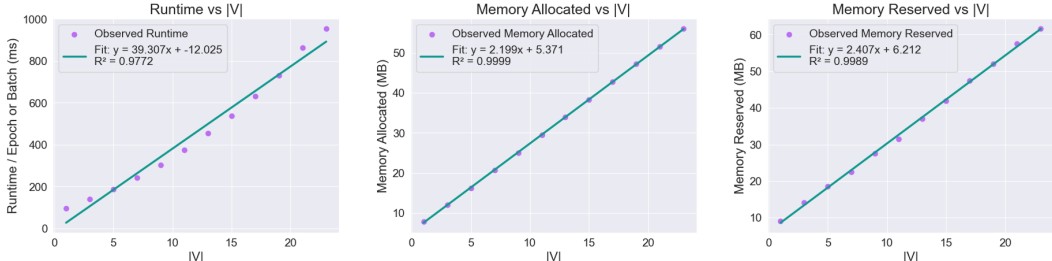

Figure 11: **Runtime and Memory usage scale linearly with size of flow generator set $|\mathbf{V}|$**. Results generated with the FERNN-$V_n^T$ on Translating MNIST, restricted to translation in only the x-direction.

# C  Variations of FERNNs

## C.1  FERNN with Non-Trivial Lift

As noted in §4, it is possible to build a FERNN with a non-trivial lift, such that the lifting convolution itself incorporates the flow transformation for each $\nu$ dimension.

Explicitly, we can define such a lift as:

$$[f_t \,\hat{\star}_{V \times G}\, \mathcal{U}^i](\nu, g) = \sum_{x \in X} \sum_{k=1}^{K} f_k(x) \mathcal{U}_k^i(g^{-1} \cdot \psi_t(\nu)^{-1} \cdot x) \tag{99}$$

Another way to think of this, is that there is a time-parameterized input kernel defined on the full space $V \times G$, i.e. $\hat{\mathcal{U}}(\nu, g, t) = \mathcal{U}(\psi_t(\nu)^{-1} \cdot g)$; however, we find this viewpoint less elegant given that the kernel then depends on time.

We see then when the flow is incorporated into the lifting convolution, the output of the convolution is no longer constant along the $\nu$ index (as it was in the trivial lift). Instead, it is now flowing according to the $\nu$'th vector field. Therefore, when processing an input undergoing a given flow $\psi(\hat{\nu})$, this input flow will combine with the flows of the lifting convolution to yield a shift along the $V$ dimensions, similar to what we previously observed in the hidden state, i.e.:

$$[(\psi_t(\hat{\nu}) \cdot f_t) \,\hat{\star}_{V \times G}\, \mathcal{U}^i](\nu, g) = \sum_{x \in X} \sum_{k=1}^{K} f_k(\psi_t(\hat{\nu})^{-1} \cdot x) \mathcal{U}_k^i(g^{-1} \cdot \psi_t(\nu)^{-1} \cdot x) \tag{100}$$

$$= \sum_{\hat{x} \in X} \sum_{k=1}^{K} f_k(\hat{x}) \mathcal{U}_k^i(g^{-1} \cdot \psi_t(\nu - \hat{\nu})^{-1} \cdot \hat{x}) \quad \text{(where } \hat{x} = \psi_t(\hat{\nu})^{-1} \cdot x)$$

$$\tag{101}$$

$$= [f_t \,\hat{\star}_{V \times G}\, \mathcal{U}^i](\nu - \hat{\nu}, g) \tag{102}$$

Notably then, keeping the flow convolution from Equation 9 unchanged, we see that we must remove the additional $\psi_1(\nu)$ shift from the original FERNN in order to maintain flow equivariance. Specifically, the new non-trivial-lift recurrence relation is then given simply as:

$$h_{t+1}(\nu, g) = \sigma\big([h_t \star_{V \times G} \mathcal{W}](\nu, g) + [f_t \,\hat{\star}_{V \times G}\, \mathcal{U}](\nu, g)\big). \tag{103}$$

In this setting, the action of the flow on the output space changes to just a permutation of the $V$ dimension, with no corresponding flow on g:

$$(\psi(\hat{\nu}) \cdot h[f])_t(\nu, g) = h_t[f](\nu - \hat{\nu}, g) \tag{104}$$

In a sense, this model can be seen as 'undoing' the action of each flow on the input when lifting. We find this to be somewhat analogous to the traditional group-equivariant CNN design choice where the transformation can *either* be applied to the filter or the input. In the FERNN setting, the 'filter' is now defined by the full recurrence relation, so we can either apply the flow transformation to the input sequence, or to the hidden state sequence.

Overall, we find this to be a slightly less elegant construction since the indexing of the kernel in the convolution is then dependent on the time index explicitly. In the trivial-lift setting introduced in the main text, this time-dependence is rather implicitly imposed by the recurrence of the hidden state itself, therefore allowing us to only require the instantaneous one-step flows during each recurrent update. Regardless, we are interested in future work which may explore this non-trivial lift setting more fully, and other interpretations of the FERNN model as described.

# D   Related Work

In this section we provide an overview of work which is related to flow equivariance and equivariance with respect to time-parameterized symmetries generally.

## D.1   Flow Equivariance without a Hidden State

As mentioned in the main text, it is possible to achieve flow equivariance without an explicitly flow-equivariant sequence model. The two primary methods for accomplishing this are through frame-wise application of an equivariant model (as described in §3) and through group-convolution over the entire space-time block (as described in §6). Both of these methods are verifiably flow-equivariant, however they are fundamentally a different class of model than what we have described in this work. They are not recurrent sequence models, and therefore intrinsically have a finite temporal context or 'receptive field' which can be used when computing any output. By contrast, recurrent networks can theoretically have an infinite temporal context, if need be, through the maintenance of a hidden-state. Given this is a fundamental distinction between recurrent and non-recurrent networks which is the subject of research beyond the domains of equivariance, we find it to be beyond the scope of this work to compare with these models explicitly. Instead, we propose flow equivariant RNNs as an extension of existing equivariant network theory to this class of models which maintain a hidden state and can operate in the online recurrent setting.

The list of prior work which can be included in this category of 'flow equivariance without a hidden state' is quite broad, since it encompasses most of the equivariant deep learning literature to date, however we list a few notable examples here. The Lorentz equivariant work of Bogatskiy et al. [2020], Gong et al. [2022] is the most relevant, while other applied work has developed 3-D convolutional networks which are equivariant with respect to Galilean shifts (our translation flows) in the context of event-based cameras [Zhu et al., 2019], or rotations over time in the context of medical imaging [Zhu et al., 2024]. Early work developed Minkowski CNNs [Choy et al., 2019], which are equivariant with respect to 4D translations, thus making them equivariant to axis-aligned motions. Further, Clifford Steerable CNNs [Zhdanov et al., 2024] have also been developed to achieve Poincaré-equivariance on Minkowski spacetime. Related work on equivariance for PDE solving / forecasting has built dynamics models which are equivariant with respect to galilean transformations in the sense that the model is equivariant if the input vector field has a global additive constant [Wang et al., 2021]. While this is valid for neural networks applied to vector field data as input, it is clearly not the same as our method in more general settings. The method of Wang et al. [2021] can be interpreted as viewing a dynamical system which has an unknown global current introduced, while ours is better interpreted as viewing the dynamical system from a moving reference frame – the two concepts are compatible and may even be combined.

## D.2   'Statically Equivariant' Sequence Models

The second broad category of related work includes sequence models which are equivariant with respect to instantaneous static group transformations, but are not equivariant with respect to time-parameterized group transformations, as our FERNN is. Examples in this category include [Azari and Erdoğmuş, 2022, Nguyen et al., 2023, Basu et al., 2023], which introduce equivariance to transformations such as static rotations in sequence models including RNNs. For example, these models train on one frame of reference and then test on a 'rotated' frame of reference. Our work can be seen to generalize these models to instead allow them to be tested on *'rotating'* frames of reference. This class of prior work can most readily be compared to the group equivariant RNN (G-RNN) we describe in §2. Other researchers have developed sequence to sequence models which are equivariant with respect to fixed permutations and demonstrated that this is beneficial in the context of language modeling [Gordon et al., 2020]. Further, recent work has developed an equivariant sequence autoencoder which uses group convolutions in an LSTM to achieve static equivariance for the purpose of PDE modeling [Fromme et al., 2025].

## D.3   Neuroscience and Biologically Inspired Neural Networks

The study of symmetry has also grown increasingly relevant in the computational neuroscience literature. Zhang et al. [2022] have studied equivariant representations in biological systems in

terms of continuous attractor networks. Interestingly, these models also use convolutional recurrent dynamics; however, again, this work only considers static translations and not motion over time. Others such as Lindeberg [2013] have developed a general theory for motion equivariant receptive fields in the visual system [Lindeberg, 2025], and even built spiking recurrent networks which integrate such filters [Pedersen et al., 2025], yet these models are equivariant only in the encoder portion of the network, not in the recurrence as we propose for FERNNs.

One class of models which is highly related to the FERNN in both theory and implementation comes from a line of biologically inspired work aiming to learn symmetries from data. One of the first models in this category, the Topographic VAE [Keller and Welling, 2022] can be seen as similar to our FERNN but with an explicitly imposed translation flow of a single velocity in the latent space. This makes these models equivariant to input transformations which are isomorphic to the translation group on the integers modulo the capsule length. Interestingly, the authors find that by simply imposing this translation flow in the latent space, the model learns to encode dataset symmetries into these flows in order to better model the dataset. This seems to imply that simply imposing flow symmetries in sequence models without a priori knowledge of the structure of the flow symmetries in the input may still be beneficial. Similar results were shown with the Neural Wave Machine [Keller and Welling, 2023], where flows in the latent space were implemented implicitly through a bias towards traveling wave dynamics. Finally, perhaps most interestingly, related work on traveling waves in simple recurrent neural networks (the wave-RNN) [Keller et al., 2024a] and SSMs [Keller, 2025] has demonstrated that waves implemented through similar 'roll' operations have significant benefits for long-term memory in recurrent neural network architectures. Incredibly, these models are identical to translation flow-equivariant RNNs in implementation, but without any mention of equivariance, and applied to an entirely different set of tasks. It is therefore of great interest to study if there is something unique to translation flows which benefit memory performance, or if similar performance benefits may be gained from any latent flow symmetry.

Finally, convolutional RNNs are becoming increasingly of interest in the computational neuroscience and 'NeuroAI' domains [Spoerer et al., 2017], with notable examples being trained as 'foundation models' for mouse visual cortex [Wang et al., 2025], and recently demonstrating state-of-the-art performance in modeling rodent somatosensory cortex [Chung et al., 2025]. Our work provides a new theoretical lens through which to study and build such models, offering potential novel insights into biological neural systems.

### D.4 Reference Frames in Neural Networks

As mentioned in §4, one way to interpret the flow-equivariant RNN is that its hidden state lives in a number of moving reference frames simultaneously (one for each $\nu \in V$). Thus, for moving inputs, the corresponding co-moving hidden state reference frame will see the input as stationary, and process it as normal. This idea of reference frames in neural networks is not new, and significant interesting related work should be noted.

Specifically, Spatial Transformer Networks [Jaderberg et al., 2016], and Recurrent Spatial Transformer Networks [Sønderby et al., 2015] can be seen to predict a frame of reference for a given input and then switch to that reference frame to gain invariance properties. However, these models do not discuss moving reference frames. Other models have been built in this vein with respect to other symmetries, such as polar coordinate networks which are inherently equivariant with respect to scale and rotation [Esteves et al., 2018]. Related in theory is the idea of Capsule Networks [Sabour et al., 2017, Hinton et al., 2018]. These models do have an explicit notion of reference frames, similar to ours, and use this to gain a structured equivariant representation, but again this is defined only in the spatial context.

### D.5 Broadly Related Work

More broadly, prior work has looked at the integration of recurrence and motion modeling specifically for vision [Wu et al., 2021, Gehrig and Scaramuzza, 2023]. These models do not have any mention of motion equivariance, and therefore are highly unlikely to provide the strong generalization benefits such as those that we present in this paper. Other work has studied ego motion for action recognition [López-Cifuentes et al., 2020]; and relevant work has looked at equivariance in the context of object tracking [Gupta et al., 2020, Sosnovik et al., 2020a].

## D.6 Equivariant Dynamical Systems

In the dynamical systems literature, equivariance is typically defined for autonomous (or homogeneous) dynamical systems which have no 'input' or 'driving force'. Abstractly, for a system $\frac{dx}{dt} = f(x)$, we say the dynamical system is equivariant if $f(g \cdot x) = g \cdot f(x)$ for all $g \in G$. Then for any $x(t)$ that solves the differential equation, we also know that $g \cdot x(t)$ solves the differential equation for the full group orbit $g \in G$ [Moehlis and Knobloch, 2007]. Similar to the equivariant neural network setting, we see that the 'output' of an equivariant dynamical system (interpreted as the particular solution) transforms in a predictable well-behaved manner for a given transformation of the input. As the simplest example, $\frac{dx}{dt} = x + x^3$, has a sign flip symmetry in $x$, meaning that if take the sign flipped version of the trajectory $x(t)$, the solution to the equation also inherits a sign flip. It is straightforward to see from the above definition that if the function $f(x)$ defining the time derivative is equivariant with respect to $G$, then the system is considered equivariant, since the definitions are equivalent.

In this work, we are interested in recurrent neural networks, which generally operate in the "forced" or non-autonomous setting. The study of symmetries in non-autonomous dynamical systems has been previously explored in the control theory literature, and has proven highly valuable for the design of robust and high performing equivariant filters [Mahony et al., 2022]. In this setting, the dynamical system is defined as $\frac{dx}{dt} = f(x, u)$ for some driving force $u$, and the equivariance property is then defined as $g \cdot f(x, u) = f(g \cdot x, g \cdot u)$. We see that our FERNNs indeed are equivariant non-autonomous dynamical systems by this definition.

The difference of our current work with this prior control-observer work, is that Mahony et al. [2022] treats time-parameterised symmetries as known biases inside a hand-written dynamical model and uses lifts/adjoint operators to keep the estimation error autonomous. The FERNN instead learns the dynamics, lifts the hidden state to a group-indexed field, and enforces equivariance by a simple group-convolution weight-sharing rule. This removes the need for handwritten dynamical models and allows learning flexible non-linear dynamics – capabilities not covered by existing observer theory.

# E  Extended Results

## E.1  In-Distribution Next-Step Prediction (Table 1 & Figure 3)

In Figure 12, we show the sequence predictions of the models presented in Figure 3 & Table 1, trained on Translating MNIST $V_2^T$ and Rotating MNIST $V_4^R$, and evaluated on the same flows. We see that when tested in-distribution, all models appear to perform well from visual inspection.

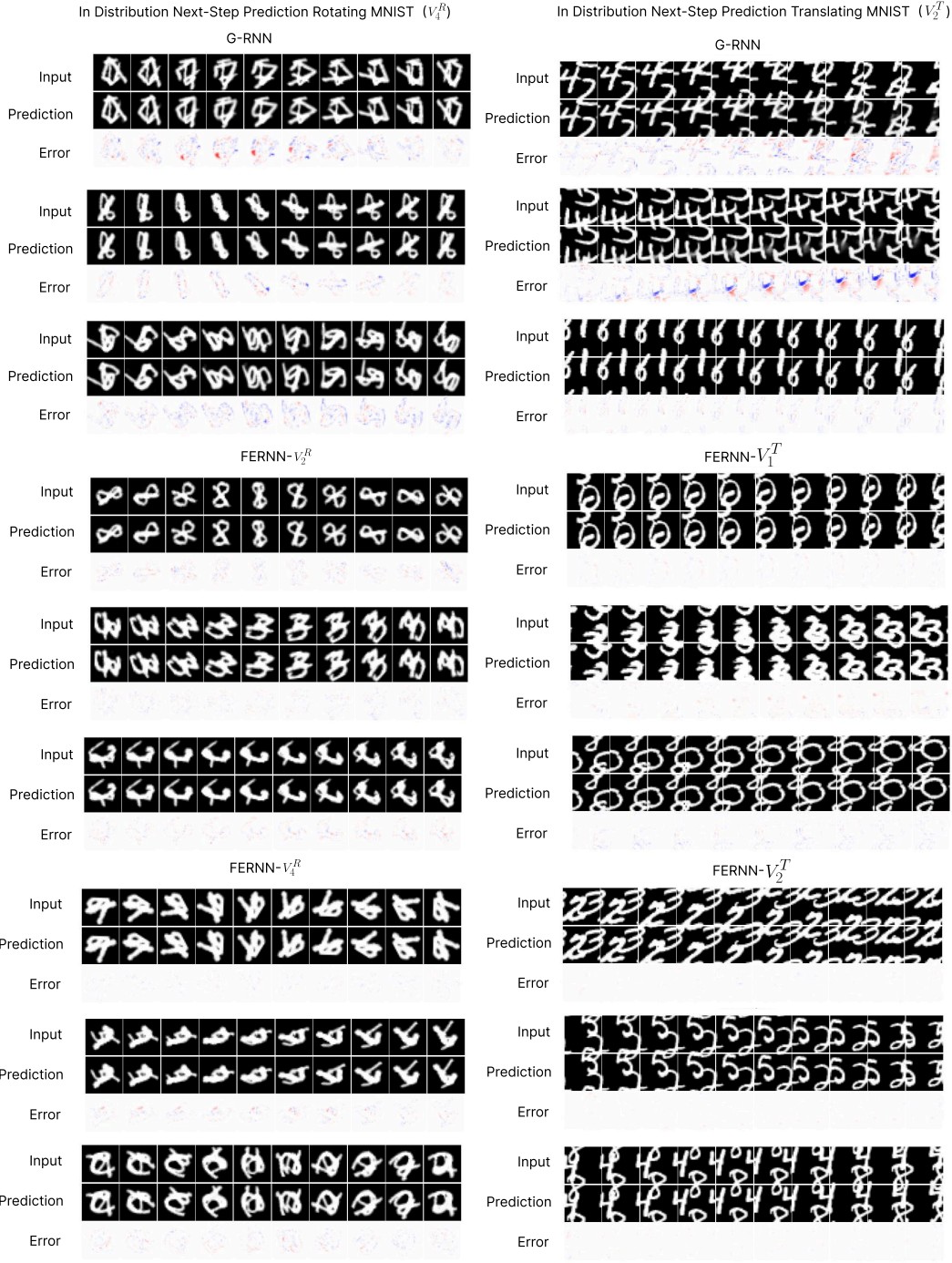

Figure 12: In-distribution sequence predictions for the models from Table 1 & Figure 3, trained on Rotating MNIST $V_4^R$ (left) and Translating MNIST $V_2^T$ (right), evaluated on the same flows.

## E.2 Length Generalization MSE vs. Forward Prediction Plot (Rotation)

In Figure 13, we show the length generalization plot, analogous plot to Figure 4 (right), but for Rotating MNIST. We see that the length generalization performance gap is not as significant on Rotating MNIST compared with Translating MNIST. We suspect that this is due to the accumulation of errors induced by repeated interpolation when performing rotation by small angles on a discrete grid. Despite this, we see that the FERNN-$V_4^R$ still achieves strong generalization up to 30-time steps forward, significantly outperforming the non-equivariant model.

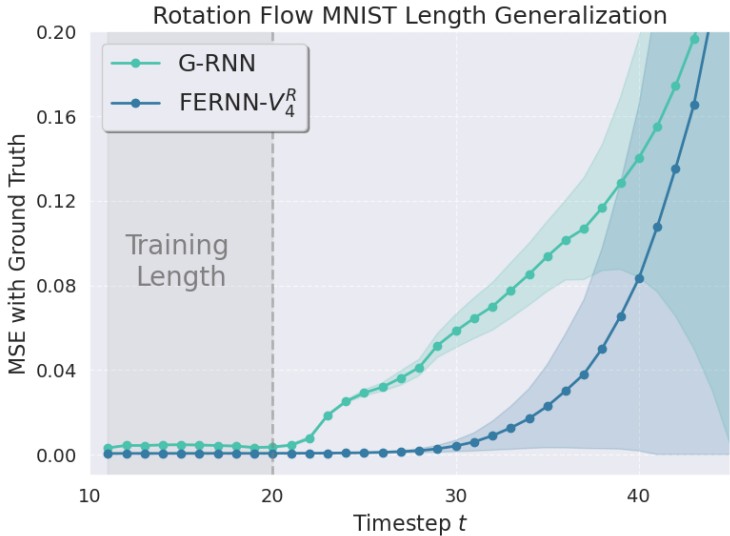

Figure 13: MSE vs. Forward prediction horizon for models on Rotating MNIST. Analogous plot to Figure 4 (right) but for Rotating MNIST.

### E.3 Length Generalization Visualizations (Translation)

In Figure 14 we show the sequence predictions of the same models presented in Table 1 & Figure 3, trained on Translating MNIST $V_2^T$ and Rotating MNIST $V_4^R$, but tested in the length generalization setting. We plot the 70-forward prediction steps here, subsampled by half in time (giving 35 elements). We plot the ground truth sequences on top, and the forward predictions below, with the error (in blue-red color scheme) on bottom. We see the FERNNs significantly outperform the G-RNNs in length generalization on Translating MNIST, mirroring the quantitative results in Figures 13 & 4.

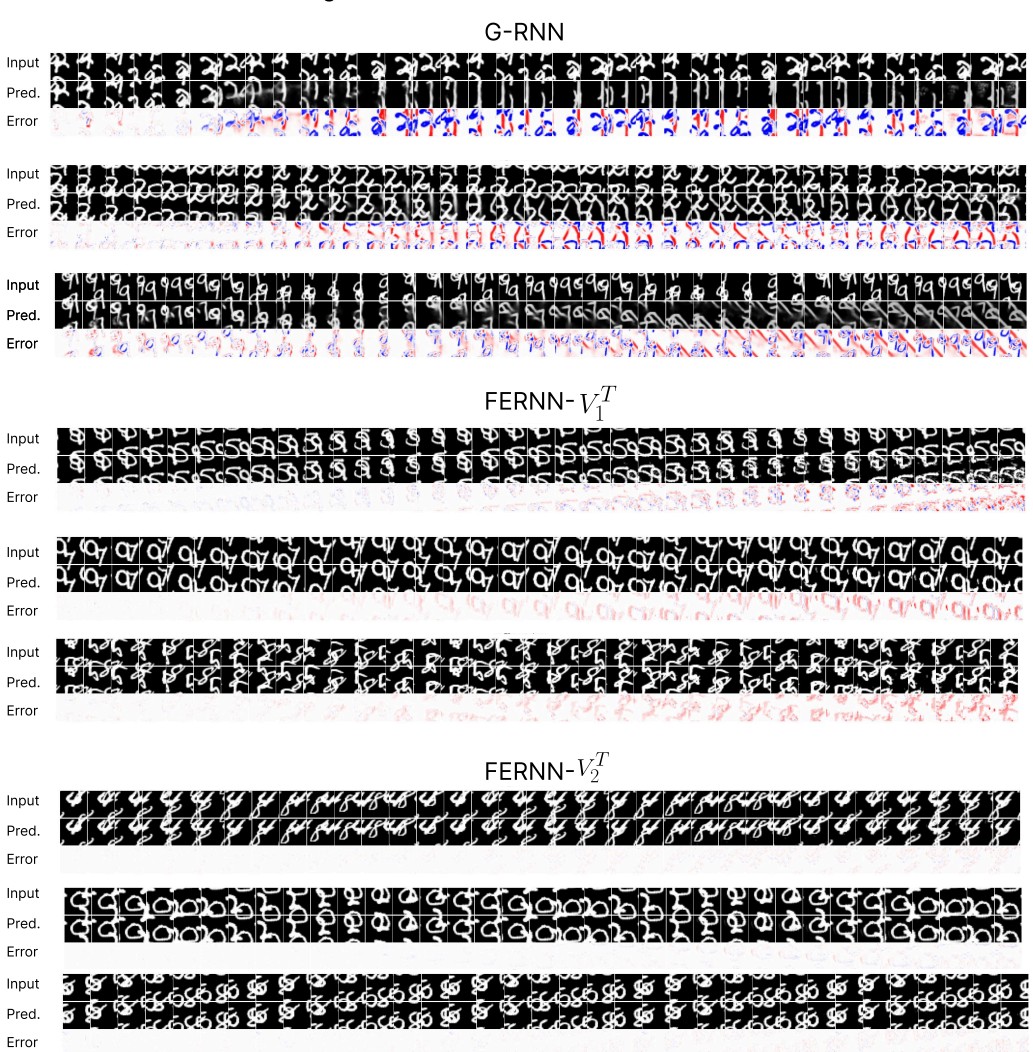

Figure 14: Samples from models trained on Translating MNIST $V_2^T$ with training sequence lengths of 20, tested on sequence lengths of 80. We plot the 70-forward prediction steps here, subsampled by half in time (giving 35 elements).

## E.4 Velocity Generalization Visualizations

In Figure 15 we show the sequence predictions of the models presented in Figure 5, trained on Translating MNIST $V_1^T$ and Rotating MNIST $V_1^R$, but evaluated on on $V_2^T$ and $V_4^R$ respectively. These figures show that FERNNs which are equivariant to flows beyond their training distribution are able to automatically generalize to these transformations at test time, achieving near perfect next-step prediction performance where non-flow equivariant RNNs fail.

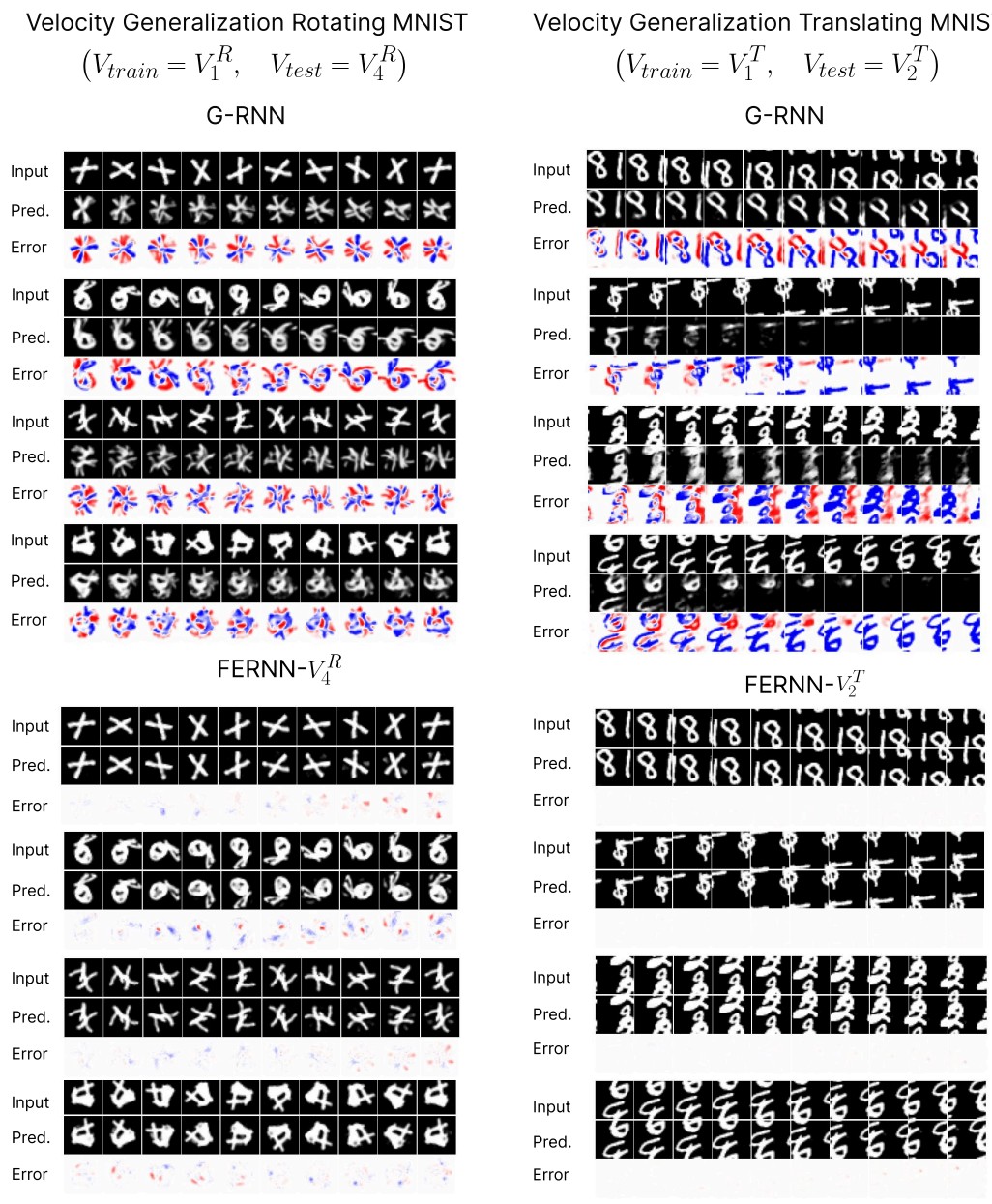

Figure 15: Samples from models trained on Rotating MNIST $V_1^R$ (left) and Translating MNIST $V_1^T$ (right), and tested on sequences with significantly higher velocity flows ($V_4^R$ and $V_2^T$ respectively). We see that the FERNN (bottom rows) has no problem generalizing to these new velocities despite never having seen them in training, while the G-RNN (top) fails. The specific flows plotted in this example are $\pm 40 \frac{deg}{step}$ for Rotating MNIST (left), and $\nu = (\pm 2, \pm 2)$ for Translating MNIST.

### E.5 Flow Equivariant RNNs and Variable Velocity Flows

By definition, flows formally maintain a constant velocity over time: they exponentiate a single constant generator element of the Lie algebra. This property is in fact necessary to get the flow composition property described as $(\psi_s(\nu) \cdot \psi_t(\nu) = \psi_{s+t}(\nu))$, and this flow composition property is essential to the equivariance proof of the FERNN (see §A.7).

For real world sequences, this can be quite limiting, since most natural data contains flows of varying velocities throughout the sequence length. However, even though the theory only extends to constant velocity flows, *in practice*, we observe that sequences with variable velocity flows are still better modeled by FERNNs than non-flow-equivariant counterparts. We provide empirical results of this on two additional non-constant-flow datasets below, and further provide intuition for why convolutional weights between lifted velocity dimensions (we call $V$-mixing) may allow for better modeling interactions between velocity channels, such as those resulting from elastic object collisions.

#### E.5.1 Variable Velocity Moving KTH Action Recognition

First, we test our model on the same moving KTH action recognition dataset as in the main text, but with a random velocity switch halfway through the sequence. Since it is entirely random, it is ultimately unpredictable to the model. Despite this, we find that the FERNN models still outperform the non-flow equivariant baselines by a significant margin, indicating the practical benefits of flow equivariance hold even when the theoretical assumptions may be violated.

| Model | Test Acc. ↑ |
|---|---|
| 3D-CNN | $0.59 \pm 0.02$ |
| G-RNN+ | $0.62 \pm 0.04$ |
| G-RNN | $0.65 \pm 0.03$ |
| FERNN-$V_1^T$ | $\mathbf{0.69 \pm 0.04}$ |
| FERNN-$V_2^T$ | $\mathbf{0.69 \pm 0.02}$ |

Table 3: **FERNNs can model variable velocity flows better than baselines.**. Test accuracy (mean $\pm$ std) for models trained and tested on the Moving KTH dataset ($V_2^T$) with random velocity changes halfway through each sequence.

#### E.5.2 Mixing Across the $V$ Dimension: Bouncing MNIST

We additionally experiment with a 'bouncing' variant of the Translating MNIST dataset. Specifically, we construct this dataset identically to the Translating MNIST dataset from the manuscript, but with the 2 digits now having elastic collisions with the image boundary and with each other. This causes digits to (predictably) change velocity at multiple points during the sequence as a function of their own velocity, and the velocity of the other objects that they collide with. Thus, to forward predict these sequences, models must be able to model interactions between velocities. In our framework, this is precisely the role of inter-$V$ terms of the flow convolution kernel of Equation 9.

In order to have space for both digits and the boundary, we increase the canvas size to $28 \times 48$. We additionally draw tight bounding boxes around the digits (in white, value $1.0$) and draw a one-pixel image boundary on the canvas (in gray, value $0.5$). Such boundaries make it easier for the model to identify collisions. In Table 4 below we report the results of the FERNN and the G-RNN baseline on this dataset, as well as the results of a FERNN with convolutional kernels that are non-zero for the cross-$V$ terms (denoted $V$-Mixing).

Ultimately, we see that the FERNN model nearly halves the error compared with the G-RNN baseline, and the addition of velocity mixing parameters further improves the performance, despite having no guarantee of equivariance with respect to these highly complex flow transformations. Intuitively, we believe $V$-mixing is most beneficial when there are dynamical computations that require the *interaction of multiple different velocity features simultaneously* to predict the next frame. In essence, one may think of the flow convolution as identifying collisions in space, and then transferring activation between opposing velocity channels to model the interaction. The constant velocity global

| Model | Bouncing MNIST Test MSE $\pm$ Std. $\downarrow$ |
|---|---|
| G-RNN | $2.5e{-}2 \pm 3e{-}4$ |
| FERNN-$V_2^T$ | $1.8e{-}2 \pm 3e{-}4$ |
| FERNN-$V_2^T$ + $V$-Mixing | $\mathbf{8.9e{-}3 \pm 9e{-}4}$ |

Table 4: **FERNNs can model elastic collisions, and $V$-mixing improves modeling flow interactions**. Mean $\pm$ std over 3 random seeds on a variant of the translating MNIST dataset where digits bounce off each other and walls elastically.

flows in the Translating MNIST task of the main text require no such inter-velocity features, and therefore we found no benefit in practice to adding $V$-mixing to the main text results.

### E.6 Moving KTH with Data Augmentation

As noted in the main text, the flowing MNIST dataset can be seen as already 'data augmented' since the velocities are randomly re-sampled for each training iteration. For the Moving KTH dataset however, this is not the case by default. In Table 5 below, we show the corresponding results for the Moving KTH dataset when re-sampling the velocities at each iteration, analogous to 'data augmentation' by camera motion.

| Model | Test Acc. $\uparrow$ | w/ Data Augmentation |
|---|---|---|
| 3D-CNN | $0.626 \pm 0.02$ | $0.742 \pm 0.01$ |
| G-RNN+ | $0.639 \pm 0.02$ | $0.662 \pm 0.04$ |
| G-RNN | $0.665 \pm 0.03$ | $0.684 \pm 0.04$ |
| FERNN-$V_1^T$ | $0.698 \pm 0.03$ | $0.694 \pm 0.05$ |
| FERNN-$V_2^T$ | $\mathbf{0.716 \pm 0.04}$ | $\mathbf{0.751 \pm 0.01}$ |

Table 5: **FERNNs still outperform non-equivariant baselines on Moving KTH dataset with data augmentation**. Test accuracy (mean $\pm$ std) for models trained and tested on the Moving KTH dataset ($V_2^T$) with velocities re-sampled at each training iteration, analogous to data augmentation. We see that all models improve performance with data augmentation as expected, and the 3D-CNN improves the most, however the FERNN still performs the best and significantly better than the comparable G-RNN non-flow-equivariant baselines.

### E.7 Larger Baselines

In our exploration we have found that increasing the number of channels of the non-equivariant baselines has led to only marginal increases in performance at best (MNIST), and significantly worse performance in other cases (KTH).

Concretely, on the KTH dataset, when multiplying the number of G-RNN channels by 25 (the number of flow generator elements, $|V|$, for FERNN-$V_2^T$), we found that we needed to reduce the learning rate from $3e{-}4$ to $1e{-}4$ to stabilize training, and even then, the model reached only $54.6 \pm 1\%$ test accuracy, under-performing the original G-RNN baseline value of $66.5\%$ test accuracy. We believe this drop in performance is due to the significantly increased number of trainable parameters resulting in increased training difficulty and more overfitting. On Translating MNIST, the amount of memory usage required to fit these additional parameters and activations prevents us from multiplying the number of channels by 25. When multiplying the number of channels by 8 (the maximum we could fit on our GPUs), we find that performance is only marginally improved from $8.1e{-}3$ (the original 128 channels) to $2.0e{-}3$ (1024 channels), however this is still significantly below the FERNN-$V_2^T$ performance of $1.5e{-}4$ (128 channels).

