# OpenReview forum: "Flow Equivariant Recurrent Neural Networks"
_NeurIPS.cc/2025/Conference — NeurIPS 2025 spotlight_

### Official Review · Reviewer_oXJm · 2025-06-12

**Clarity:** 1
**Significance:** 3
**Originality:** 4
**Rating:** 5
**Confidence:** 2

**Summary:**

This paper introduces a novel approach to integrate continuous, structured transformations, referred to as "flows" (like motion), into neural networks designed for sequential data. The authors define these flows as one-parameter Lie subgroups. They define "flow equivariance" and demonstrate that existing sequence models, even those built with components already equivariant to static symmetries, do not inherently possess this flow equivariance. The paper then presents their "Flow Equivariant Recurrent Neural Network (FERNN)" model , which adapts the RNN structure by incorporating principles from Group Equivariant Convolutional Neural Networks (G-CNNs) to achieve flow equivariance. Empirically, the authors show that FERNNs offer significant advantages in terms of training speed, generalization capabilities, and performance.

**Questions:**

* Line 84 - “..$(g\cdot \Phi)(y)$..” f is missing?
* Equation 3 - $\mathcal{W}^i_k$ appears in both convolution equations in Equation (3). However, its domain seems to change. in the lifting convolution as a function of $X$ and in the group convolution as a function of $G$. While this might be consistent if $G=\mathbb{Z}^2\subset \mathbb{R}^2$ it's not generally well-defined for arbitrary domains $X$ and $G$.  Also, The superscript i is not defined in $\mathcal{W}^i_k$  and same goes for the subscript $f_k$.
* Regarding the model in Equation (3), a question arises: shouldn't there be an additional operation to project the features from the group space to some output space? This operation would likely resemble the lifting convolution.
* Line 130-131 $f$ is missing from the equivariance definition?

**Ethical Concerns:**

["NO or VERY MINOR ethics concerns only"]

**Final Justification:**

The authors have successfully addressed my concerns about the paper's readability. Additionally, they conducted extensive new experiments to further validate their claims. For these reasons, I will keep my current rating of accept.

**Limitations:**

yes

**Paper Formatting Concerns:**

no issues

**Quality:**

3

**Strengths And Weaknesses:**

Strengths -
* This paper presents a novel solution to a new problem: handling data with flow symmetry, a challenge that has not been extensively studied. The authors propose a unique approach and demonstrate why existing methods are inadequate for this specific problem.

Weaknesses -
* The paper's heavy reliance on notations and complex formulations, while understandable given its theoretical depth, makes it challenging to follow. Although simplifying intricate theoretical concepts can be difficult, the inclusion of a running example would significantly aid in motivating the proposed method and clarifying the distinction between static and flow transformations, a point that Figure 1 does not adequately address.

---

> ### Author Rebuttal · Authors · 2025-07-31
>
> ### **Summary**
>
> We thank the reviewer for their time spent with our manuscript. We appreciate that they find our work to be *novel* in both goals and methods, agreeing with sentiments of the other reviewers.
>
> The reviewer has suggested numerous areas for improvement of the clarity of our work, which we have taken to heart. Below we propose how we aim implement these clarifying modifications to our manuscript.
>
> ### **Clarity**
>
> We sincerely thank the reviewer for their suggestions about how to improve the clarity of our submission. We readily admit that in pursuit of technical precision, and to match the conventions of prior equivariance literature, the manuscript has become less accessible to those outside the field. We believe this complication is not strictly necessary, and therefore appreciate the Reviewer encouraging us strive for greater clarity.
>
> To facilitate communication of the main ideas and findings of our work, we plan to change the following:
> - We plan to add a visualization of the difference between static equivariance and flow equivariance to the main text.
> - We plan to add an additional visualization of the representation of the flow action in the FERNN latent space for the simple bump example (visualizing the result of the main Theorem, 4.1).
> - We plan to add visualizations of data flows to the main text (on KTH), for a more immediate conceptual picture of what a flow symmetry is.
> - Finally, as reviewer vWda has suggested, our detailed review of G-equivariance can be shortened and moved to the appendix to reduce extended mathematical complication of the main text.
>
> We additionally plan to release a blog post accompanying the paper which will have the following clarifying properties:
> - a motivational introduction to the 'flow equivariance problem'.
> - a direct visual and motivational comparison between static equivariance and flow equivariance.
> - an extension of the 'bump' example which runs throughout the blog post, showing the difference between the G-RNN and FERNN, and the visualization of the flow action.
>
> We propose to include a link to this blog post in the paper to assist those who stumble on the paper but prefer a gentler introduction.
>
>
> ### **Questions**
> - The reviewer is correct that $f$ is missing in line 84. It should be:  $(g \cdot \Phi [f])(y) := \rho_Y(g) \Phi [f] (\pi_Y(g^{-1}) y)$. We appreciate them mentioning this error.
> - We appreciate the reviewer bringing up where we missed defining indices $i$ and $k$ in Equation 3. These are the output and input channels for the convolution respectively, following the original group convolution notation of Cohen & Welling (2016). We have made this explicit in the camera ready draft, and have further made explicit the domains of the lifting and group convolutions, as the reviewer noted this may have been ill defined.
> - With respect to output mappings, this choice is typically left up to the modeler, and is not part of traditional equivariance frameworks. Again we defer to the formulation of Cohen & Welling (2016), of which our Equation 3 is merely a restatement.
> - In lines 130-131 the reviewer is correct that $f$ is again missing. It should be: $(\psi (\nu) \cdot \Phi  [f] )_t(y) := \Phi_t [f] ( \psi_t ( \nu )^{-1} \cdot y)$. We greatly appreciate the reviewer catching these notational inconsistencies and we have amended them in the camera ready draft.
>
> We warmly welcome discussion on any further points the reviewer may have in regards to how we may improve the clarity of the text.

---

> > ### Comment · Reviewer_oXJm · 2025-08-03
> >
> > I appreciate the authors' thorough response and for addressing all the points raised. Making the paper more accessible would be valuable, as it would enable a broader range of researchers to engage with the intriguing problem it presents.

---

### Official Review · Reviewer_scMk · 2025-07-02

**Clarity:** 3
**Significance:** 2
**Originality:** 3
**Rating:** 5
**Confidence:** 4

**Summary:**

This paper introduced the concept of flow equivariance for sequence models where sequence data undergo flow actions defined by a set of generators and a time variable t. The paper discussed why recurrent neural networks equivariant to single-frame transformations are not flow-equivariant and derived flow-equivariant RNNs based on lifting and truncated group convolutions. Experiments are conducted on a flowing MNIST dataset and an action recognition dataset with artificial translation flows, and they show the effectiveness of flow-equivariance.

**Questions:**

1. Experiments on data with less restrictive assumptions. For example, with a time-varying flow or a flow that lands in between the discretized generators.

2. Computational overhead from the introduced flow equivariance.

**Ethical Concerns:**

["NO or VERY MINOR ethics concerns only"]

**Final Justification:**

My concerns in the review are resolved. I recommend accepting this paper.

**Limitations:**

Yes

**Quality:**

2

**Strengths And Weaknesses:**

Strengths:
The paper proposed a new type of equivariance that addresses the symmetry of sequential data with flows from specific generators. Good originality here. The notations, definitions, and derivations are overall clear. The visualization, especially figure 1 with a simplified example, helps convey the key idea.

Weaknesses:
The designed network FERNN only works if the flow is generated from a predefined discrete set of generators, which is limiting, and not the case in most practical scenarios. The experiments are restricted with the following unrealistic assumptions: 1. the flow is spatially global; 2. the flow is of constant speed; 3. the flow generators are a discrete set small enough to be covered in the lifted group dimension. I think that at least the second condition can be relaxed for the proposed model. The authors could do experiments on data with non-constant flow. Overall, the flow model used in this paper is overly simplified, and it is hard for me to see its practical value.

The paper did not disclose the memory usage and computational resources needed for the models. Given the group convolution nature, I would anticipate significant overhead. Could you compare the FERNN, G-RNN, and regular RNNs?

---

> ### Author Rebuttal · Authors · 2025-07-31
>
> ### **Summary**
> We sincerely thank the Reviewer scMK for their time spent with our work, and for greatly helping us to improve the manuscript. We appreciate that they commend the *originality*, *clarity* and *exposition* of our work.
>
> In our response, we directly address the Reviewer's primary concern about the practical applicability of our method. This concern appears to be due to the perception that the benefits of our proposed model are limited to *spatially global flows*, *constant speed flows*, and *flows which exactly match the model's small discrete set of generators*.
>
> Through a suite of new results, we demonstrate that the *practical benefits* of FERNNs are not so limited, even if the theory does not yet extend this far. Explicitly, we present positive empirical results on:
> 1. Non-global flows,
> 2. Variable-velocity flows,
> 3. Data flows landing between the model's generator set.
>
> In each of these cases, we demonstrate FERNNs still excel over non-flow-equivariant baselines.
>
> We subsequently include a quantitative analysis of the computational overhead of flow-equivariance, both in terms of memory and compute time, and how this scales with the size of the generator set $V$ (equivalent to a notion of the 'amount of flow equivariance').
>
> Finally, we conclude with a comparison of the limitations raised by the Reviewer and those present in the original Group Equivariant Convolutional Neural Networks (G-CNNs) of Cohen & Welling (2016). We highlight that G-CNNs have delivered remarkable practical value since their introduction (as noted by Reviewer vWda) despite very similar limitations, and thus suggest that our work may be interpreted analogously as the introduction of a new foundational framework for sequence-model equivariance, opening the door for future work to address the theoretical limitations.
>
> ### **Non-Global Flows**
> To demonstrate that the benefits of FERNNs extend to non-spatially global flows, we provide new results on a 2-Digit Flowing MNIST task. The task is identical to that in the main paper, except that 2 random digits are simultaneously placed on the canvas, and each digit is 'flowed' independently. Visually, this looks like two digits moving in two separate random directions (for translation), or two overlapping digits rotating at different angular velocities (for rotation). The full image sequence is thus not describable by a single global flow (in the formal mathematical sense).
>
> | Model | 2-Digit Translating MNIST (Test MSE) | 2-Digit Rotating MNIST (Test MSE) |
> | ------- | ----------- | -------- |
> | G-RNN | 8.1e−3 $\pm$ 6e−4 | 4.0e−3 $\pm$ 5e−4 |
> | FERNN-$V_1^T$/$V_2^R$ | 5.3e−4 $\pm$ 8e−5 | 1.3e−3 $\pm$ 5e−5 |
> | FERNN-$V_2^T$/$V_4^R$ | **1.5e−4 $\pm$ 2e−5** | **6.1e−4 $\pm$ 3e−5** |
>
> As can be seen, FERNNs still achieve roughly an order of magnitude lower forward prediction error than their non-flow-equivariant counterparts. This strong performance is explained by the fact that while each $\nu$ channel flows globally, each pixel is simultaneously represented in the full lifted space of all $\nu \in V$. Therefore, the readout from the hidden state (in our case, max-pooling) can select an independent velocity channel to represent each spatial location.
>
> ### **Variable-Velocity Flows**
> The reviewer is correct to note that *constant speed is inherent to the mathematical definition of a flow*: flows exponentiate a single constant generator element of the Lie algebra. This property is in fact necessary to get the flow composition property described on line 125 ($\psi_s(\nu) \cdot \psi_t(\nu) = \psi_{s+t}(\nu)$), and this flow composition property is essential to the equivariance proof of the FERNN (see line 230, & Eqn. 83 of Supplement).
>
> However, even though the theory only extends to constant velocity flows, *in practice*, we observe that sequences with variable velocity flows are still better modeled by FERNNs than non-flow-equivariant counterparts. We provide tables below demonstrating this for two non-constant-flow datasets:
>
> **'Bouncing MNIST', where 2-digits bounce off each other and the image boundary.**
>
> | Model| MNIST Test MSE $\pm$ std. |
> | -------- | ---------- |
> | G-RNN | 2.5e-2 $\pm$ 3.2e-4       |
> | FERNN-$V_2^T$ | 1.8e-2 $\pm$ 3.1e-4       |
> | FERNN-$V_2^T$ + V-Mixing | **8.9e-3 $\pm$ 9.3e-4**   |
>
>
> **KTH with a random velocity switch halfway through the sequence.**
>
> | Model | KTH Test Acc $\pm$ std. |
> | ------------- | ----------------------- |
> | 3D-CNN        | 0.59 $\pm$ 0.016        |
> | G-RNN+        | 0.62 $\pm$ 0.043        |
> | G-RNN         | 0.65 $\pm$ 0.034        |
> | FERNN-$V_1^T$ | **0.69 $\pm$ 0.036**    |
> | FERNN-$V_2^T$ | **0.69 $\pm$ 0.017**    |
>
> We see that on Bouncing MNIST, the FERNN model nearly halves the error, and the addition of velocity mixing features further improves the performance. We refer the Reviewer to our reply to Reviewer vWda for an extended discussion $V$-mixing and the dataset details. We similarly see on the variable velocity KTH dataset, FERNNs again significantly outperform their non-flow-equivariant counterparts.
>
> ### **Handling Data Flows Outside the Model's Generator Set $V$**
> Finally, to demonstrate that FERNNs yield improved performance even when data 'lands between' the elements of the generator set, we generate a Translating MNIST dataset with x-velocity flows from -5 to 5 pix/step (0 y-velocity), and train models with equivariance to subsets of these data-generating flows.
>
> In the table below, we report the test set MSE (mean over 3 seeds) computed for each x-velocity separately (Avg. at end). We see that *any* flow equivariance improves the average loss over non-flow-equivariant models, and further, when the data lands between the elements of the generator set (bolded), the FERNNs always achieve lower error than the G-RNN, sometimes by up to an order of magnitude, despite not being analytically equivariant to these flows. In the camera ready draft, we plan to include a figure of this data to improve readability.
>
>
> | Model | -5       | -4       | -3       | -2       | -1       | 0    | 1        | 2        | 3        | 4        | 5        | Avg  |
> | ----------- | -------- | -------- | -------- | -------- | -------- | ---- | -------- | -------- | -------- | -------- | -------- | ---- |
> | G-RNN | **1e-3** | **9e-4** | **1e-3** | **1e-3** | **7e-4** | 5e-4 | **6e-4** | **1e-3** | **1e-3** | **1e-3** | **1e-3** | 9e-4 |
> | FERNN-$V_{-1, 0, 1}^T$ | **7e-4** | **7e-4** | **7e-4** | **5e-4** | 3e-4     | 3e-4 | 4e-4     | **6e-4** | **7e-4** | **7e-4** | **7e-4** | 6e-4 |
> | FERNN-$V_{-5, 0, 5}^T$ | 2e-4     | **4e-4** | **9e-4** | **8e-4** | **4e-4** | 2e-4 | **4e-4** | **8e-4** | **1e-3** | **5e-4** | 2e-4     | 5e-4 |
> | FERNN-$V_{-3, 0, 3}^T$ | **8e-4** | **2e-4** | 3e-4     | **2e-4** | **2e-4** | 2e-4 | **2e-4** | **2e-4** | 3e-4     | **2e-4** | **8e-4** | 3e-4 |
> | FERNN-$V_{-5, -3, -1, 0, 1, 3, 5}^T$ | 9e-5     | **1e-4** | 8e-5     | **1e-4** | 6e-5     | 6e-5 | 8e-5     | **1e-4** | 8e-5     | **1e-4** | 8e-5     | 9e-5 |
> | FERNN-$V_{[-5, 5]}^T$ | 5e-5     | 5e-5     | 5e-5     | 5e-5     | 5e-5     | 4e-5 | 5e-5     | 5e-5     | 5e-5     | 5e-5     | 5e-5     | 5e-5 |
>
>
> ### **Computational Overhead**
> Finally, we study the computational overhead of flow equivariance. Theoretically, both compute and memory should scale approximately linearly with the number of flow generators (denoted $\|V\|$). In the tables below, we validate this for FERNNs with different sized generator sets. Note the G-RNN is equivalent to a FERNN with $\|V\| = 1$, i.e. $V = ${$0$}.
>
> | Model | Memory Allocated (GB) | Runtime per Epoch (s) |
> | -------- | -------- | ----- |
> | G-RNN ($\|V\| = 1$) | 7.9 | 96.9 |
> | FERNN $\|V\| = 5$   | 16.3 | 186.8 |
> | FERNN $\|V\| = 9$   | 25.1 | 304.1 |
> | FERNN $\|V\| = 15$  | 38.3 | 537.9 |
> | FERNN $\|V\| = 19$  | 47.2 | 731.6 |
>
> If we fit a linear model to these results, we obtain extremely good fits, and the following parameters:
>
> | Metric  | Intercept | Slope (per $V$ element) | R² (goodness‑of‑fit) |
> | ------- | --------- | --------- | --------- |
> | Memory  | 5.4 GB    | + 2.2 GB | 0.99 |
> | Runtime | ‑12.0 s   | + 39.3 s | 0.98 |
>
> We note that our current FERNN implementation is extremely computationally inefficient (looping over each flow generator), while a scan-based optimized implementation of this model is certainly possible and would dramatically reduce runtime and improve its scalability.
>
> ### **Conclusion**
> In conclusion, we thank the reviewer for encouraging these new experiments. We believe many readers are likely to share the same concerns, so we are grateful for the opportunity to demonstrate empirically that these limitations are likely not as severe as theory alone might suggest.
>
> Interestingly, the limitations brought up by the reviewer are nearly identical to those of the original Group Equivariant Neural Networks of Cohen & Welling (2016) -- namely, *scalability,* *discrete group equivariance* and *the restriction to global transformations*. With time, however, researchers have developed methods to ameliorate these limitations: such as steerable equivariance (Weiler et al. 2019), addressing the discrete approximation limitation, and simultaneously scalability; and gauge equivariance (Cohen et al. 2019), addressing the local vs. global transformation limitation. In doing so, as Reviewer vWda has pointed out, group-equivariance has become incredibly practically relevant, including being used in highly popularized models such as AlphaFold 2.
>
> We therefore suggest that our paper may be seen as a the first step in the introduction of a new theoretical foundation -- namely a foundation for dynamic equivariance in sequence models that opens the door for future work. We welcome further discussion.

---

> > ### Comment · Reviewer_scMk · 2025-08-03
> >
> > Thanks for adding the new results along with the discussions. I think the empirical benefits under the cases beyond the assumptions of the equivariant model are reasonable. However, what if the non-equivariant model is scaled to a similar size as the equivariant models? For example, augmenting the feature dimension comparable to |V|?
> >
> > Besides, is the linear fit to the runtime wrong? It does not match the column of runtime per epoch in the table above.

---

> > > ### Author Response · Authors · 2025-08-05
> > >
> > > We thank the reviewer for reading our rebuttal. We appreciate that they agree with our finding that the empirical benefits of the model extend beyond the limitations of the assumptions; hopefully alleviating their concerns about the practical value of the model.
> > >
> > > With respect to the reviewer's two additional questions:
> > >
> > > ### **Equivalent Sized Baselines**
> > > The reviewer asks how the performance of the Flow Equivariant RNN compares with a non-equivariant RNN (G-RNN) with an equivalent number of channels (e.g. multiplied by the size of the equivariant generator set |V|). To answer succinctly, in our exploration we have found that increasing the number of channels of the non-equivariant baselines has led to only marginal increases in performance at best (MNIST), and significantly worse performance in other cases (KTH).
> > >
> > > Concretely, on the KTH dataset, when multiplying the number of G-RNN channels by 25 (the number of flow generator elements, |$V$|, for FERNN-$V_2^T$), we found that we needed to reduce the learning rate from 3e-4 to 1e-4 to stabilize training, and even then, the model reached only $54.6 \pm 1$% test accuracy, under-performing the original G-RNN baseline value of $66.5$% test accuracy. We believe this drop in performance is due to the significantly increased number of trainable parameters resulting in increased training difficulty and more overfitting.
> > >
> > > On 2-Digit Translating MNIST, the amount of memory usage required to fit these additional parameters and activations prevents us from multiplying the number of channels by 25. When multiplying the number of channels by 8 (the maximum we could fit on our GPUs), we find that performance is only marginally improved from 8.1e-3 (the original 128 channels) to 2.0e-3 (1024 channels), however this is still significantly worse than the FERNN-$V_2^T$ performance of 1.5e-4 (128 channels).
> > >
> > > In addition to this relatively poor test set performance, the increased channels have no additional structure to them, and therefore theoretically provide no guarantees with respect to velocity generalization or length generalization. While we have not explored the generalization capability of these larger models explicitly, we are confident that they will not demonstrate the same generalization benefits that we display in the main paper. We will happily include these additional studies in the final draft.
> > >
> > > Finally, we note that in order to circumvent the limitations above, and provide another comparable baseline, we have already included the additional 'G-RNN+' baseline in the main text for the KTH datasets. This baseline has no additional parameters, making it easier to train, but has the same number of activations through the same lifting process, where the hidden states are propagated forward by a learned convolutional filter, rather than the prescribed Lie Algebra flows. What we see is that this model performs significantly worse than the exactly flow equivariant models, highlighting that simply including additional hidden state dimensions with max-pooling does not explain the performance benefits of the FERNN.
> > >
> > > ### **Runtime Linear Fit**
> > > The reviewer asks about the quality of the linear fit of runtime vs. |$V$| (size of the generator set $V$). We thank the reviewer for allowing us to expand on this question, because we could not fit the full table in the earlier response due to character length limits.
> > >
> > > We include the full table we used to compute the linear fits below, along with the values of the fit, bolded next to the corresponding metric. We can see that, as the reviewer noted, while the fit is not perfect, sometimes above or below the line, the runtime value is clearly linear in the size of $V$ up to very large values of $|V|$, and thus the $R^2$ value is indeed 0.979. In the camera ready version of the paper, we will add a plot of these values and the linear fit to aid in interpretability.
> > >
> > > | Model               | Memory Allocated | Memory Fit | Runtime / Epoch | Runtime Fit |
> > > | ---- | ----- | ---- | ----- | --- |
> > > | G-RNN ($\|V\| = 1$) | 7.86 | **7.62**   | 96.827 | **27.3**    |
> > > | FERNN $\|V\| = 3$   | 11.98 | **12.06**  | 140.103 | **105.9**   |
> > > | FERNN $\|V\| = 5$   | 16.26 | **16.5**   | 186.769 | **184.5**   |
> > > | FERNN $\|V\| = 7$   | 20.65 | **20.94**  | 241.451 | **263.1**   |
> > > | FERNN $\|V\| = 9$   | 25.07 | **25.38**  | 304.133 | **341.7**   |
> > > | FERNN $\|V\| = 11$  | 29.48 | **29.82**  | 375.115 | **420.3**   |
> > > | FERNN $\|V\| = 13$  | 33.89 | **34.26**  | 453.314 | **498.9**   |
> > > | FERNN $\|V\| = 15$  | 38.32 | **38.7**   | 537.857 | **577.5**   |
> > > | FERNN $\|V\| = 17$  | 42.76 | **43.14**  | 630.887 | **656.1**   |
> > > | FERNN $\|V\| = 19$  | 47.18 | **47.58**  | 731.624 | **734.7**   |
> > > | FERNN $\|V\| = 21$  | 51.61 | **52.02**  | 863.686 | **813.3**   |
> > > | FERNN $\|V\| = 23$  | 56.04 | **56.46**  | 954.190 | **891.9**   |
> > >
> > > We thank the reviewer again for their time and welcome any further questions.

---

> > > > ### Comment · Reviewer_scMk · 2025-08-08
> > > >
> > > > Thank you. My concerns are resolved.

---

### Official Review · Reviewer_ngxM · 2025-07-03

**Clarity:** 4
**Significance:** 4
**Originality:** 4
**Rating:** 6
**Confidence:** 3

**Summary:**

This paper proposes, analyzes, and proposes a model architecture for flow equivariance. Flow equivariance is symmetry that exists in the context of time series prediction where a transformation on the input of a function produce the same result as the same transformation to the output of the function. Examples of flow equivariant processes are next step prediction models and generalization over flow velocities. This paper states and analyzes flow equivariance and proposes a flow equivariant convolution and recurrence relation. The resulting FERNN (flow equivariant recurrent neural networks) are shown to have the desired equivariance properties and to improve performance on simple one-step and sequence prediction tasks performed on a "flowing MNIST" dataset.

**Questions:**

none

**Ethical Concerns:**

["NO or VERY MINOR ethics concerns only"]

**Final Justification:**

I think this is a strong paper. I think the main strengths are: 1) flow equivariance seems to be a timely and important problem with a range of applications. I also like that this is an intuitive idea at its core. 2) The mathematical grounding and analysis here is terrific.

The biggest weakness in my view is that the various MNIST experiments still feel like toy examples. However, I nevertheless think the strengths outweigh this weakness.

**Limitations:**

See my comments in strengths and weaknesses. It would be a major step to demonstrate this in the context of more challenging flow prediction problems.

**Quality:**

4

**Strengths And Weaknesses:**

Strengths:

-- Flow equivariance seems to be a timely and important problem. The applications are significant and the approach formalizes a fundamental intuitive idea.

-- The analysis is terrific. The g-rnn counter example is great. The mathematical development is right on target.

-- The experiments bear out the claims of the paper. Indeed flow equivariance seems to help in sequence prediction tasks, at least in the simple "flowing MNIST" case considered.


Weaknesses:

-- The experimental domains (flowing MNIST) are very simple. It would be a big step to demonstrate this on slightly more complex flow prediction problems.

-- It is unclear what are the computational requirements of the FERNN layers. Apparently, eq 9 does a convolution in time as well as space. This strikes me as potentially expensive. It would be great to understand this better and discuss.

---

> ### Author Rebuttal · Authors · 2025-07-31
>
> ### **Summary**
> We sincerely thank Reviewer ngxM for their strong support of our work. We appreciate that they acknowledge the *timeliness* and *importance* of introducing equivariance for sequence models at this point in time. We are very grateful that they find our analysis and exposition exceptional, and that they find our claims to be well supported by our experiments.
>
> In our response, we will address the two weaknesses listed by the reviewer: namely, *simplicity of the modeled data flows*, and *the computational complexity of the FERNN model*.
>
> Specifically, we first highlight a number of more complex flow prediction problems that we have tested FERNNs on since our submission of the original manuscript. Secondly, we clarify the computational requirements of FERNNs both theoretically and empirically.
>
> ### **More Complex Flow Prediction**
> Below we present results for three new sequence modeling tasks with more complex flow symmetries as part of the data generating process. We see in each case that the flow equivariant models maintain their advantage over non-flow-equivariant counterparts.
>
> #### **2-Digit Independently Flowing MNIST**
> The first dataset we explore is a 2-digit variant of the flowing MNIST task. The task is identical to that in the main paper, except that 2 random digits are simultaneously placed on the canvas, and each digit is 'flowed' according to its own independent randomly sampled flow generator. Visually, this looks like two digits moving in two separate random directions (for translation), or two overlapping digits rotating at different angular velocities (for rotation). Below we report the forward prediction test set MSE in the same setting as in the main paper (10 steps of input, forward predict the next 10 steps).
>
> | Model                 | 2-Digit Translating MNIST (Test MSE) | 2-Digit Rotating MNIST (Test MSE) |
> | --------------------- | ------------------------------------ | --------------------------------- |
> | G-RNN                 | 8.1e−3 ± 6e−4                        | 4.0e−3 ± 5e−4                     |
> | FERNN-$V_1^T$/$V_2^R$ | 5.3e−4 ± 8e−5                        | 1.3e−3 ± 5e−5                     |
> | FERNN-$V_2^T$/$V_4^R$ | **1.5e−4 ± 2e−5**                    | **6.1e−4 ± 3e−5**                 |
>
>  Although it is not apparent from this table, we have also explored the length generalization and velocity generalization of the FERNN on this 2-digit MNIST task and we find that it performs equally as well as for the 1-digit task, demonstrating that all of the benefits of flow equivariance appear to function as expected in this more complex flow setting.
>
> While still a toy dataset, this modification represents a significant new ability of the model -- namely, the ability to model multiple input flows simultaneously. One can imagine this as a highly simplified version of the visual transformations that would occur when one simultaneously combines camera motion with the independent motion of the video subject: while the camera motion may induce a global flow in one direction, the subject is free to move in an entirely independent direction (e.g. a camera pans to the left while a basketball player runs up on the screen). We are pleased to show that FERNNs can handle this type of scenario.
>
>
> #### **KTH with a random velocity switch halfway through the sequence**
> The second more complex flow type we explore is non-constant-velocity flows. Specifically we explore the ability of FERNNs to model sequences where the flow velocity switches to a new random velocity mid-way through the sequence. We test this on the KTH action recognition dataset by modifying our simulated 'camera motion' with such a switch. We again see that FERNNs outperform their non-flow-equivariant counterparts significantly.
>
> | Model         | KTH Test Acc $\pm$ std. |
> | ------------- | ----------------------- |
> | 3D-CNN        | 0.59 $\pm$ 0.016        |
> | G-RNN+        | 0.62 $\pm$ 0.043        |
> | G-RNN         | 0.65 $\pm$ 0.034        |
> | FERNN-$V_1^T$ | **0.69 $\pm$ 0.036**    |
> | FERNN-$V_2^T$ | **0.69 $\pm$ 0.017**    |
>
>
> #### **'Bouncing MNIST', where 2-digits bounce off each other and the image boundary**
> The third and perhaps most complex dataset we experiment with is a 'bouncing' variant of the flowing MNIST dataset. Specifically, we construct this dataset identically to the Translating MNIST dataset from the manuscript, but with 2 digits which now have elastic collisions with the image boundary and with each-other. This causes digits to change velocity at multiple points during the sequence as a function of their own velocity, and the velocity of the other objects that they collide with. Thus to forward predict these sequences, models must be good at modeling interactions between velocities.
>
> In order to have space for both digits and the boundary, we increase the canvas size to 28x48. We additionally draw tight bounding boxes around the digits (in white, value 1.0) and draw a one-pixel image boundary on the canvas (in gray, value 0.5). In the table below we report the results of the FERNN and the G-RNN baseline on this dataset. We additionally report the results of a FERNN with convolutional kernels that are non-zero for the cross-$V$ terms (denoted $V$-Mixing). We refer the Reviewer to our discussion with Reviewer vWda for a more in-depth discussion of the role of $V$-mixing, and why it helps here.
>
> | Model                    | MNIST Test MSE $\pm$ std. |
> | ------------------------ | ------------------------- |
> | G-RNN                    | 2.5e-2 $\pm$ 3.2e-4       |
> | FERNN-$V_2^T$            | 1.8e-2 $\pm$ 3.1e-4       |
> | FERNN-$V_2^T$ + V-Mixing | **8.9e-3 $\pm$ 9.3e-4**   |
>
> Ultimately, we see that  FERNNs again outperform the non-flow-equivariant counterparts on this dataset, despite having no guarantee of equivariance with respect to these highly complex flow transformations. We find the fact that $V$-mixing improves *modeling of object interactions* to be very exciting, and a direction certainly worth exploring further in future work.
>
> ### **Computational Requirements of FERNN layers**
> Finally, we address the computational complexity of the FERNN layer. The flow convolution in Equation 9 performs a convolution over standard group elements ($g \in G$, which can be thought of as spatial positions for the simple translation case), and also over flow generators ($\nu \in V$, which can be thought of as movement velocities). This can be implemented efficiently as a 3D convolution for the case of 2D images, and a 1D generator group -- (as we do in the above model denoted $V$-mixing). For sets with higher dimensional generators, one can use N-D convolutions, which are also implemented efficiently in frameworks like Jax.
>
> The computational complexity of the model scales linearly with the size of the set of generators $V$ (which we denote $|V|$). Below we report the memory requirements and runtime per epoch as a function of the size of this set.  Note the G-RNN is equivalent to a FERNN with $|V| = 1$, i.e. $V = $ {$0$}.
>
> | Model | Memory Allocated (GB) | Runtime / Epoch (s) |
> | ------------------- | --------------------- | ------------------- |
> | G-RNN ($\|V\| = 1$) | 7.9 | 96.9 |
> | FERNN $\|V\| = 5$   | 16.3 | 186.8 |
> | FERNN $\|V\| = 9$   | 25.1 | 304.1 |
> | FERNN $\|V\| = 15$  | 38.3 | 537.9 |
> | FERNN $\|V\| = 19$  | 47.2 | 731.6 |
>
> If we fit a linear model to these results, we obtain extremely good fits, and the following parameters:
>
> | Metric  | Intercept | Slope (per $V$ element) | R² (goodness‑of‑fit) |
> | ------- | --------- | ----------------------- | -------------------- |
> | Memory  | 5.4 GB    | + 2.2 GB                | 0.99                 |
> | Runtime | ‑12.0 s   | + 39.3 s                | 0.98                 |
>
> We note that our current FERNN implementation is extremely computationally inefficient (looping over each flow generator), while a scan-based optimized implementation of this model is certainly possible and would dramatically reduce runtime and improve its scalability.  In our limitations and future work section, we describe how something like this may be possible within the State Space Model (SSM) framework.
>
> ### **Conclusion**
> We thank the reviewer for their engagement with our work and strong support. We welcome further discussion of any of the above points.

---

### Official Review · Reviewer_vWda · 2025-07-06

**Clarity:** 3
**Significance:** 3
**Originality:** 3
**Rating:** 5
**Confidence:** 4

**Summary:**

This paper presents recurrent neural networks that are equivariant w.r.t. given time-parameterized symmetry transformations on the input sequence, referred to as flows. For example, for a sequence of images, a symmetry transformation $\psi: \N, \X \to \X$ could be such that $\psi(t, x_t)$ translates input image $x_t$ by $t$ pixels. To achieve this, they use the same idea as in group convolutional networks, which is to lift the input space from $\X$ to $\X \times G$, and in the case of recurrent networks, also lifting the hidden state from $\Z$ to $\Z \times G$, where $G$ is the symmetry group. Experimentally, they show good performance in next step prediction for MNIST digits with constant motion and also for action classification in videos with constant camera motion.

$$
\renewcommand{\N}{\mathbb{N}}
\newcommand{\X}{\mathcal{X}}
\newcommand{\Z}{\mathcal{Z}}
$$

**Questions:**

None.

**Ethical Concerns:**

["NO or VERY MINOR ethics concerns only"]

**Final Justification:**

The authors have implemented all of the changes that I suggested for improving the paper. I think the paper is currently fit for publication at neurips so I recommend accept.

**Limitations:**

Some of the weakness that I've described could be added to the limitations.

**Paper Formatting Concerns:**

Minor issue: some figures are not readable when printed. See for example the heatmap on the bottom right in Fig. 4. If accepted, the authors should use the extra space to make the images larger so that the text inside is legible.

**Quality:**

3

**Strengths And Weaknesses:**

## Strengths

* I like the topic of the paper and the problem that it studies (obtaining equivariance w.r.t. time-parameterized symmetries). I think it addresses an interesting under-explored topic. In this regard it is original as far as I am aware.
* The paper seems technically sound. I did not read all the proofs in detail but I did not notice any issues in the equations, claims, or proof sketches.
* The proposed solution makes sense and is connected to the idea from group convolutional layers [Cohen & Welling, 2016] for which there are a lot of research papers and positive empirical results.

## Weaknesses

* The authors mention that some of the experiments do not perform mixing across $G$ channels. Without mixing we will have a independent processing path for each $g$ and the model becomes equivalent to a kind of group averaging method, which is not the method proposed in this paper. In other words, mixing is a *core* component of the group convolution approach. Therefore, it is crucial to have some mixing. Group averaging methods should be included as a baseline (i.e., zero mixing).
* G-equivariance is an orthogonal idea and it overcomplicates the notation and exposition. Saying that G-equivariance is different from time-parameterized equivariance is necessary and useful for the paper but, afterward, it should not be included in the equations for simplicity. If the authors like, they can move the more general setup to the appendix. This is a suggestion to make the paper digestible for a wider audience and not a major issue in my opinion.
* There should be some pictures of the action recognition dataset after camera motion has been applied.
* The authors motivate their method with camera motion but, in reality, camera motion is not a constant motion; it changes direction sometimes. If direction of motion is allowed to change at every step then the size of the symmetry group will be exponentially large and the proposed method won't work. This issue can perhaps be left for future works.
* Data augmentation should also be included as a natural baseline and it can actually more closely match natural camera motion with changing directions. It is likely that most videos are centered and data augmentation would fix that. For the experiments in this paper it suffices to do augmentation with constant camera motion.

The most important issue I think is the mixing which invalidates the experiments. If one weakness is to be addressed it should be that one. I hope the authors spend some time thinking about this issue and address it and that they don't dismiss it. I'm also quite sure the numbers will improve. If the authors worry about computational difficulties, they can only add mixing to a fraction of *convolution* channels (not $G$ channels).

Summary of things to do to improve paper:
- Data augmentation baseline.
- Group averaging baseline (no mixing).
- Add some mixing across $G$.
- Expand on limitations.

---

> ### Author Rebuttal · Authors · 2025-07-31
>
> ### **Summary**
> We thank Reviewer vWda for their comments, suggestions, and critical analysis of our work. We appreciate that they find our work *original*, *technically sound*, and that it *addresses an under explored topic*. We also appreciate that they acknowledge the connection of our work to Group Convolution [Cohen & Welling, 2016], supporting our belief that FERNNs lay the theoretical groundwork for exploiting a new type of time-dependent symmetric structure in data, just as Group Convolution did for 'static' symmetric structure.
>
> Below, we reply to a few of the reviewer's comments and include results that directly address their suggested improvements regarding:
> - *Mixing across velocity ($V$) channels*
> - *Data augmentation baselines*
> - *Realistic non-constant camera motion*
>
> As a high level summary of our response:
> 1. We provide new results with mixing across $V$ channels for both the original MNIST data (where we see no improvement) and a new non-constant velocity 'bouncing MNIST' dataset (where we see improvement), and explain these results with a mechanistic hypothesis for the functional role of $V$-mixing.
> 2. We clarify that our baselines on MNIST can already be understood as 'data augmented' from our dataset construction procedure. We provide new results on the KTH action recognition dataset with proper data augmentation and still find the FERNN performs best.
> 3. We provide new results on both MNIST and KTH with non-constant velocity flows. We see that the FERNN still outperforms non-flow-equivariant baselines, despite the mismatch with theory, supporting the practical relevance of our approach.
>
>
> ### **Mixing Across Velocity ($V$) Channels**
>
> We thank the reviewer for highlighting the importance of mixing across $V$ channels, and encouraging us to add these experiments; we find the results quite illuminating.
>
> In the table below we show the performance (Test MSE $\pm$ std. over 3 seeds) of the G-RNN, original FERNN, and FERNN with $V$-mixing (as described in Eqn. 9, with a convolutional kernel of size 3 in the $V$ dimension). We find that on the original Translating MNIST dataset (first column), the results are not significantly different with $V$-Mixing added, and the model actually performs measurably worse. As mentioned in the main text (lines 248-255), the reason for not including $V$-mixing in our initial results is not due to computational limitations, but rather a conscious design choice to limit the propagation of equivariance error arising from the truncation of the infinite velocity group to a finite range. This is similar to what was found in prior work on scale-equivariance [Worall and Welling, 2019, Sosnovik et al., 2020b], and is likely the reason for the drop in performance.
>
> However, we believe we can also understand why $V$-mixing *does not improve* performance significantly here -- namely, we believe $V$-mixing is most beneficial when there are dynamical computations that require the *interaction of multiple different velocity features simultaneously* to predict the next frame. The constant velocity global flows in Translating MNIST require no such features.
>
> To test this mechanistic hypothesis for the role of $V$-mixing, we provide new results on a dataset we call 'bouncing MNIST', identical to translating MNIST, but where MNIST digits now have elastic collisions with the image boundary and with each-other (we have now added two digits per image, and increased the image size to 28x48 correspondingly). These collisions require exactly the types of muli-velocity computations we believe $V$-mixing would help with. Experimentally, we validate this hypothesis in the second column of the table below, where the FERNN with $V$-Mixing achieves the lowest test error on this bouncing MNIST dataset.
>
> | Model                    | Translating MNIST (MSE) | Bouncing MNIST (MSE)          |
> | ------------------------ | ----------------------- | ----------------------- |
> | G-RNN                    | 1.4e-3 $\pm$ 8.9e-6     | 2.5e-2 $\pm$ 3.2e-4     |
> | FERNN-$V_2^T$            | **1.8e-4 $\pm$ 2.6e-6** | 1.8e-2 $\pm$ 3.1e-4     |
> | FERNN-$V_2^T$ + V-Mixing | 3.0e-4 $\pm$ 2.8e-5     | **8.9e-3 $\pm$ 9.3e-4** |
>
> ### **Data Augmentation**
> With respect to data augmentation, we appreciate the reviewer's suggestion; we include a general clarification of our data generation procedure below, followed by new results for the KTH data (both of which we will add to the main text).
>
> **Clarification:**
> Since the datasets we use in our paper have no natural motion, they are already 'augmented' in some way in order to generate the training sequences. The main design choice is then whether the velocities ('motions') are sampled once for each image and fixed throughout training, or if velocities are re-sampled at each epoch (each time the image is presented). This second option, we call 're-sampling', is more akin to standard data augmentation.
>
> **MNIST results are already 'data augmented' according to the 're-sampling' definition:**
> We highlight that the MNIST results in our paper are already 'data augmented' according to the 'resampling' definition above. This is because the velocities are randomly sampled for each training example in the data loader by default, meaning each digit is observed with a different velocity at each epoch, equivalent to data augmentation. This is described briefly on lines 270-272, and more thoroughly in Supplementary Material Section B.2. The only setting where this augmentation is not applied is on the velocity generalization tasks, where it would not make sense by definition of zero-shot generalization.
>
> **KTH results are not 're-sampling' data augmented, new results below:**
> The KTH (action recognition) results in Table 2 are not with 're-sampled' augmentations for each epoch, instead they are sampled once (with a fixed seed for all models) at the beginning of training. We appreciate the reviewer bringing attention to this distinction. In the table below we show the new results for Table 2 replicated with 're-sampled' data augmentation for each model. We see that the performance of all models improves, and the performance gap between models reduces, although the FERNN still outperforms the baselines by a narrow margin.
>
> | Model         | Test Acc (No Aug) | Test Acc + Data Aug   |
> | ------------- | ----------------- | --------------------- |
> | 3D-CNN        | 0.626 ± 0.02      | 0.742 $\pm$ 0.015     |
> | G-RNN+        | 0.639 ± 0.02      | 0.662 $\pm$ 0.037     |
> | G-RNN         | 0.665 ± 0.03      | 0.684 $\pm$ 0.041     |
> | FERNN-$V_1^T$ | 0.698 ± 0.03      | 0.694 $\pm$ 0.049     |
> | FERNN-$V_2^T$ | **0.716 ± 0.04**  | **0.751 $\pm$ 0.009** |
>
> In the camera ready version, we propose to include 're-sampled' data augmentation baselines in Figure 5 as well. We also propose to include non-augmented baselines on MNIST for completeness.
>
> ### **'Realistic' Non-Constant Camera Motion**
>
> The reviewer comments that natural camera motion is not a constant velocity, and suggests that this may make the method less applicable to real-world data. We agree, and further believe that the study of the performance of FERNNs on non-constant velocity flows is certainly of interest to us as well. As a preliminary step in this direction, we first appeal to the results above on 'Bouncing MNIST' where we see that the FERNN still achieves a significant performance improvement over non-flow-equivariant counterparts, despite the variable velocity flows.
>
> Secondly, we provide new results on the KTH dataset where we simulate camera motion that changes its velocity vector randomly half-way through the sequence. We see in the table below that the FERNN still significantly outperforms the baselines.
>
> | Model         | KTH Variable Velocity, Test Acc $\pm$ std. |
> | ------------- | ------------------------------------------ |
> | 3D-CNN        | 0.59 $\pm$ 0.016                           |
> | G-RNN+        | 0.62 $\pm$ 0.043                           |
> | G-RNN         | 0.65 $\pm$ 0.034                           |
> | FERNN-$V_1^T$ | **0.69 $\pm$ 0.036**                       |
> | FERNN-$V_2^T$ | **0.69 $\pm$ 0.017**                       |
>
> In future work, we believe that it may be possible to train a 'flow predictor' to estimate the current flow between any two time-steps, and apply only this flow to this hidden state. This would avoid the 'exponential growth' problem that the reviewer mentions, while still maintaining equivariance. We leave this as a potential future direction which is motivated and supported by the theory of our proposed work.
>
> ### **Response to minor points:**
> - The reviewer requested pictures of the action recognition dataset with camera motion applied. We bring to the reviewer's attention that these are indeed already in Section B.7, Figure 6, of the supplementary material. We will move these to the main paper for the camera ready.
> - In our camera ready draft we have split Figure 4 into two separate figures and removed empty space, significantly improving readability, and addressing the Reviewer's minor formatting concerns.
> - In our camera ready we have expanded on the limitations of our model with respect to variable velocity flows, data augmentation, and computational scaling.
> - We sincerely appreciate the authors suggestion to help simplify our exposition by reducing background on G-Equivariance. We agree that the paper could be simplified to improve the exposition, and reducing background mathematical clutter is a good option to do so. We do however believe we need to include a minimal example of G-equivariance in the text since it is explicitly required as a subset of a flow-equivariant network, i.e. it is not possible to build a flow-equivariant model which is not also G-equivariant.
>
> We thank the reviewer again for their suggestions and we welcome further discussion if we may have misunderstood any of their comments.

---

> > ### Comment · Reviewer_vWda · 2025-08-04
> >
> > I thank the authors for addressing the weaknesses and answering my questions. I will adjust my score accordingly.

---

### Decision · Program_Chairs · 2025-09-17

**Decision:**

Accept (spotlight)

**Comment:**

This paper introduces Flow Equivariant Recurrent Neural Networks (FERNNs), extending equivariant network theory to handle time-parameterized transformations like motion in sequential data. Four reviewers provided consistent positive evaluations, acknowledging the novel theoretical contribution while raising concerns about experimental simplicity, practical applicability given restrictive assumptions about discrete generator sets and constant flows, computational overhead, and mathematical presentation clarity. The authors effectively addressed these concerns through comprehensive rebuttals including new experiments on variable velocity flows and bouncing scenarios that demonstrate benefits extend beyond theoretical assumptions, detailed computational scaling analysis, and commitments to improve presentation with visualizations and simplified exposition. While the work has limitations—requiring predefined discrete generators and theoretical guarantees only for constant velocity flows—it represents a solid foundational contribution that opens new research directions in dynamic equivariance, drawing appropriate parallels to early group equivariant CNNs that evolved from initially restrictive frameworks into practically valuable tools. The mathematical framework is rigorous, experimental validation adequately establishes core claims within scope, and the potential for inspiring follow-up research justifies acceptance, contingent on implementing the promised clarity improvements in the camera-ready version.